

# Ice and Mixed-Phase Cloud Statistics on Antarctic Plateau

William Cossich[1,*], Tiziano Maestri[1,*], Davide Magurno[1,*], Michele Martinazzo[1], Gianluca Di Natale[2], Luca Palchetti[2], Giovanni Bianchini[2], and Massimo Del Guasta[2]

[1]Physics and Astronomy Department, Alma Mater Studiorum - University of Bologna (I)
[2]Istituto Nazionale di Ottica, Consiglio Nazionale delle Ricerche (I)
[*]These authors contributed equally to this work.

**Correspondence:** tiziano.maestri@unibo.it

**Abstract.** Statistics on the occurrence of clear skies, ice and mixed-phase clouds over the Concordia station, in the Antarctic Plateau, are provided for multiple time scales and analysed in relation to simultaneous meteorological parameters measured at the surface. Results are obtained by applying a machine learning cloud identification and classification code (named CIC) to 4 years of measurements between 2012-2105 of down-welling high spectral resolution radiances, measured by the Radiation

Explorer in the Far Infrared-Prototype for Applications and Development (REFIR-PAD) spectroradiometer. The CIC algorithm is optimized for Antarctic sky conditions (clear sky, ice clouds, and mixed-phase clouds) and results in a total hit rate of almost 0.98, where 1.0 is a perfect score. Scene truth is provided by LiDAR measurements that are concurrent with REFIR-PAD. The CIC approach demonstrates the key role of far infrared spectral measurements for clear/cloud discrimination and for cloud phase classification. Mean annual occurrences are 72.3%, 24.9% and 2.7% for clear sky, ice and mixed-phase clouds

respectively, with an inter-annual variability of a few percent. The seasonal occurrence of clear sky shows a minimum in winter (66.8%) and maxima (75-76%) during intermediate seasons. In winter the mean surface temperature is about 9°C colder in clear conditions than when ice clouds are present. Mixed-phase clouds are observed only in the warm season; in summer they amount to more than one third of total observed clouds. Their occurrence is correlated with warmer surface temperatures. In the austral summer, the mean surface air temperature is about 5°C warmer when clouds are present than in clear sky conditions. This

difference is larger during the night than in daylight hours, likely due to increased solar warming. A comparison of monthly mean results with cloud occurrence/fraction derived from gridded (Level-3) satellite products, from both passive and active sensors, emphasizes the difficulty of adequately inferring cloud/clear-sky properties in the Antarctic region and highlights the ability of the CIC/REFIR-PAD synergy to identify multiple cloud conditions and study their variability at different time scales.

## 1  Introduction

The polar regions present several challenges for meteorology and climatology studies (Walsh et al., 2018). These regions are crucial components of the Earth's Radiation Budget (ERB) (Liou, 2002; Kiehl and Trenberth, 1997), since they generally emit more energy to space in the form of infrared radiation than what is absorbed from sunlight, thereby behaving as heat sinks. Modelling studies have shown that changes in cloud properties (e.g., cloud amount, cloud thermodynamic phase, cloud height, cloud optical thickness) over Antarctica may impact different regions in the globe, highlighting the importance of Antarctic





clouds for the global climate system (Lubin et al., 1998). However, obtaining measurements of cloud properties in the interior

of the Antarctic continent is still a challenge (Town et al., 2005; Lachlan-Cope, 2010; Bromwich et al., 2012). Observations

from synoptic weather stations require an experienced observing staff and sometimes become unavailable during 'white-out'

conditions caused by blowing snow. Analysis of satellite measurements from both active and passive sensors must account

for a number of problems in inferring the cloud properties. One issue is that the cloud radiative properties tend to be very

similar to those of the background (the snow/ice surface). Optically thin cirrus clouds are often present in the Antarctic Plateau

(King and Turner, 1997) but are difficult to identify and analyse due to their small cloud signals (the difference between cloudy

and clear-sky radiances). Measurements become problematic during the long polar night (King and Turner, 1997), and some

stations reduce the observing frequency in the winter time (Bromwich et al., 2012). Observations at solar wavelengths are not

available for about half of the year, thus reducing the overall ability to recognize the presence of cloud layers and to derive

their physical and optical features. Measurements at longer wavelengths (i.e. in the infrared, or IR) are available regardless of

solar illumination but it is often the case that the cloud top temperature is similar to the ice surface temperature (King et al.,

1992; King and Turner, 1997; Bromwich et al., 2012).

Active remote sensing techniques have been very helpful in overcoming the limitations of the passive instruments in polar

regions. Adhikari et al. (2012) investigated the seasonal and interannual variabilities of the vertical and horizontal cloud distri-

butions over the southern high latitudes poleward of 60°S, using observations from CloudSat and Cloud-Aerosol LiDAR and

Infrared Pathfinder Satellite Observation (CALIPSO) satellites between June 2006 and May 2010. They found that the Antarc-

tic Plateau has the lowest cloud occurrence of the Antarctic continent ($< 30\%$). The sensors on board of the aforementioned

satellites have been also used to investigate macro and microphysical Antarctic cloud properties (Verlinden et al., 2011; Ad-

hikari et al., 2012; Listowski et al., 2019; Ricaud et al., 2020). Nevertheless, satellite active sensors are not lacking in problems

when used for cloud detection in Polar regions. For example, Chan and Comiso (2011, 2013) discuss the difficulties encoun-

tered by either the Cloud Profiling Radar (CPR), on Cloudsat, and the Cloud-Aerosol LiDAR with Orthogonal Polarization

(CALIOP), on CALIPSO, in detecting low-level clouds in the Arctic. The difficulties arise from the CloudSat coarse vertical

resolution (about 500 m) and its limited sensitivity (low signal-to-noise ratio) near the surface, and in the case of CALIOP

are due to the geometrically thin nature of the cloud and its surface proximity. Bromwich et al. (2012) present a review on

Antarctic tropospheric clouds. They discuss the instruments and methods to observe Antarctic clouds and the current data sets

available. The authors highlight that there are relatively few measurements of clouds in the Antarctic, especially in the interior.

They also indicate that better and more frequent remote sensing and *in situ* observations are needed.

The selection of the FORUM project (Palchetti et al., 2020) in 2019 as the $9^{th}$ Earth Explorer Mission by European Space

Agency (ESA) has revitalized studies in the Far-Infrared (FIR) part of the spectrum, approximately covering the 100-700 cm$^{-1}$

band. Many studies have shown that the FIR can be used to complement standard remote sensing measurements performed

in the Mid Infrared (MIR) and improve cloud detection, classification, and inference of cloud properties (Rathke et al., 2002;

Palchetti et al., 2016; Di Natale et al., 2017; Maestri et al., 2019a). Moreover, ground-based remote sensing spectral upwards-

looking measurements are very useful to determine the cloud properties relevant to the energy budget (Mahesh et al., 2001;

Cox et al., 2014; Di Natale et al., 2020).





This study is performed in this context and exploits a unique dataset derived from FIR and MIR downwelling spectral radiances measured at the Concordia station, Dome C, in the middle of the Antarctic Plateau. The measurements are performed by means of the Radiation Explorer in the Far Infrared-Prototype for Applications and Development (REFIR-PAD) Fourier transform spectroradiometer (Bianchini et al., 2019), in the scope of the projects Radiative Properties of Water Vapor and Clouds in Antarctica (PRANA) and Concordia Multi-Process Atmospheric Studies (CoMPASs), within the Italian National Program for
Research in Antarctica (PNRA, Palchetti et al., 2015). These projects represent the first long-term field campaigns to collect high spectral resolution radiances in the FIR, with continuity for an extended period (measurements started in 2012). REFIR-PAD is installed inside an insulated shelter, named the *Physics Shelter*, together with a backscattering LiDAR (acronym of Light Detection and Ranging). The LiDAR detects backscattering and depolarization signals up to 7 km above the surface. Besides these measurements, the Antarctic Meteo-Climatological Observatory installed at Concordia (http://www.climantartide.it/, last
access: $25^{th}$ January, 2021) provides data from an automatic weather station (AWS), and from daily radiosondes launches. These measurements are analyzed and correlated to the meteorological conditions observed at the Concordia station and considered representative of a large area of East Antarctica because of the horizontal uniformity in the Antarctic Plateau.

Recently, Maestri et al. (2019b) presented an algorithm to identify and classify clouds based on principal component analysis of IR radiance spectra at high spectral resolution. The Cloud Identification and Classification (CIC) is a fast machine learning
algorithm able to perform a cloud detection and classification, exploiting spectral variations of IR radiance signals. CIC can account for spectral radiance from the full IR spectrum including the MIR and FIR. The algorithm analyses a distribution of the so-called *similarity index*, that is a parameter defining the level of closeness between the analysed spectra and the elements of specific classes that are defined with training sets.

In this study, the CIC algorithm is applied to REFIR-PAD downwelling radiances to detect and classify Antarctic clouds
between 2012 and 2015. The main goal of this effort is to obtain statistics on clear/cloud occurrence and in the investigation of the diurnal cycle and seasonality of clouds in the Antarctic Plateau. Both ice and mixed-phase clouds have been considered, the latter consisting of a geometrically thin supercooled liquid water layer that, in general, may have ice particles present either above or below this liquid layer.

The algorithm is first applied to a test set so that the CIC performances can be assessed. The excellent classification scores
obtained in the testing phase provide a solid base for the application of the CIC to the entire dataset. In this study, an effort is made to link the meteorological state of the atmosphere to the cloud occurrence.

The paper is organized as follows. Section 2 describes the instrumentation and measurements performed at Concordia Station. Section 3 introduces the CIC algorithm, its set-up and optimization to identify and classify clouds. Section 4 discusses the cloud occurrences results in different time scales. The study is summarized in Section 5.

## 2   Instrumentation and Measurements

Concordia station is an Antarctic research base located at Dome C over the Antarctic Plateau (75°06'S, 123°23'E, 3.230 m AMSL), in the East Antarctic region (Figure 1). The station opened in 2005 as part of an international cooperation project

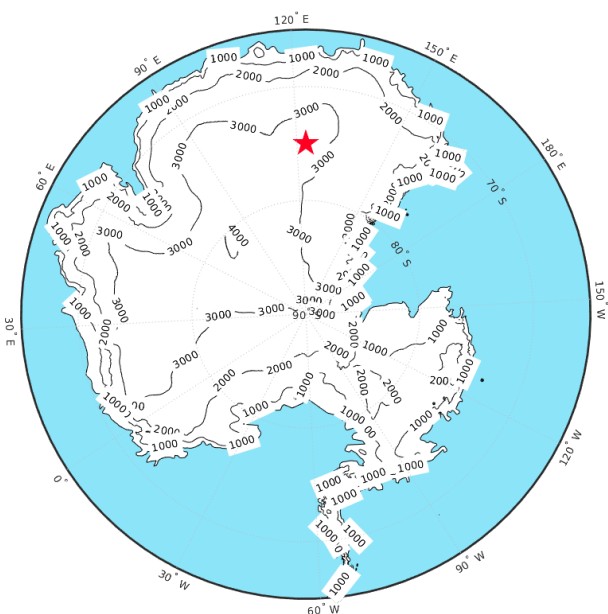

**Figure 1.** Antarctica elevation map, with the Concordia station indicated by the red star.

between the PNRA and the French Polar Institute Paul-Émile Victor (IPEV). A detailed description of the instrumentation available in the PRANA and CoMPASs experiments at Concordia station is given in Palchetti et al. (2015). A brief overview

of the instruments and measurements made between 2012 and 2015 is provided in what follows.

Spectral measurements of the downwelling radiance are performed by REFIR-PAD, which provides spectrally resolved zenith-looking radiance measurements in the range 100-1500 cm$^{-1}$ with a 0.4 cm$^{-1}$ spectral resolution, thus covering a large part of the atmospheric longwave emission including both the FIR and part of the MIR region. The instrument points at the zenith through a 1.5 m chimney. The measurement sequence to obtain one complete spectrum is made of four calibration

acquisitions, in which the instrument looks at the internal reference blackbody sources, and four sky observations. Each single acquisition takes about 80s. The entire sequence has a duration of about 14 min: 5.5 minutes of sky observations, 5.5 minutes of calibrations, and delays for detector settling after scene changes (Palchetti et al., 2015). REFIR-PAD is a fast scanning spectroradiometer with signals acquired in the time domain and resampled in postprocessing at equal intervals in optical path difference. It has been designed to operate with uncooled detectors and optics. The instrument operates full time, alternating

cycles of 5–6 hours of measurements, with 1–3 hours of analysis. It is installed in the Physics Shelter, located 500 m southward from the main station, in what is called the *Clean Air Area*, where the predominant winds keep the air clean from the exhaust plume of the Concordia power generator. Between the years 2012 and 2015, a total of 87960 spectra were analysed. The spectra annual distribution is reported in Table 1.

Atmospheric backscattering and depolarization ratio profiles are measured from a LiDAR system every 5 minutes (Palchetti

et al., 2015). This instrument operates at 532 nm and uses a Quantel laser (Brio). The measurements are made in the range

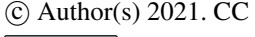



**Table 1.** Number of analysed REFIR-PAD spectra for each year.

| Year: | 2012 | 2013 | 2014 | 2015 |
|---|---|---|---|---|
| Spectra: | 16177 | 19298 | 25089 | 27396 |

30-7000 m above the surface, with 7.5 m of vertical resolution. The LiDAR telescope has refractive optics with 10 cm diameter and 30 cm focal length. There is an interference bandpass filter of 0.15 nm bandwidth. The line of sight is zenith-looking through a window, enabling measurements in all-weather conditions.

A set of REFIR-PAD spectra (1928) is co-located with LiDAR measurements. The co-location criterion is defined by the time

of measurements: each REFIR-PAD spectrum is associated to the LiDAR data that is closest in time. Co-located measurements are used to visually classify the REFIR-PAD spectra. This process requires a human intervention for the analysis of each co-located measurement and is not routinely applicable to the whole database. For these cases, cloud layers are detected from the analysis of the backscatter profiles and the depolarization ratio is used to determine the thermodynamic phase of the particles. The visually classified spectra are then used to set up training and test sets as described in more detail in the next section. In

the Antarctic environment the determination of cloud thermodynamic phase is not trivial. According to Liou and Yang (2016), liquid water droplets retain the polarization state of the incident energy, while the light beam backscattered from non-spherical ice particles is partially depolarized as a result of internal reflections and the transformation of coordinate systems governing the electric vector. A theoretical analysis performed by the same authors shows that in presence of a liquid water cloud the depolarization remains at about 2-4%, whereas radiation backscattered from non-spherical ice particles is strongly depolarized,

varying between 30 and 40%. However, the threshold to determine the water physical state in real clouds can vary depending on the atmosphere and the cloud microphysical parameters. Sassen and yu Hsueh (1998) evaluate ground-based LiDAR data in presence of contrail cirri, during the Subsonic Aircraft: Contrail and Cloud Effects Special Study (SUCCESS) field campaign. They found depolarization ratios in persisting contrails ranging from about 0.3 to 0.7. Freudenthaler et al. (1996) observed depolarization ratios of 0.1 to 0.5 for contrails with temperatures ranging from $-60$ to $-50°C$, depending on the stage of their

growth. In this study a depolarization ratio of 0.15 is used as a threshold for the discrimination of the liquid water clouds and ice clouds over the Concordia Station.

An example for the observation of a clear sky (red triangle), ice cloud (blue triangle), and mixed-phase cloud (green triangle) is provided for the LiDAR backscatter and depolarization ratio in the upper and middle panels of Figure 2. The lower panel of the same figure provides the correspondent REFIR-PAD spectra. In clear sky conditions, the LiDAR backscattering signal

decreases with altitude, while the signal increases in the presence of cloud particles. As shown in the figure, clouds can be composed of multiple layers, each one with different depolarization features. When the depolarization ratio is higher than 15%, the cloud is classified as an ice cloud (blue triangle). For lower values of the depolarization ratio, the cloud is assumed to be a mixed-phase cloud (green triangle). In particular, it is frequently observed that mixed-phase clouds are composed of a layer of liquid water near the cloud base and an ice layer at the cloud top.



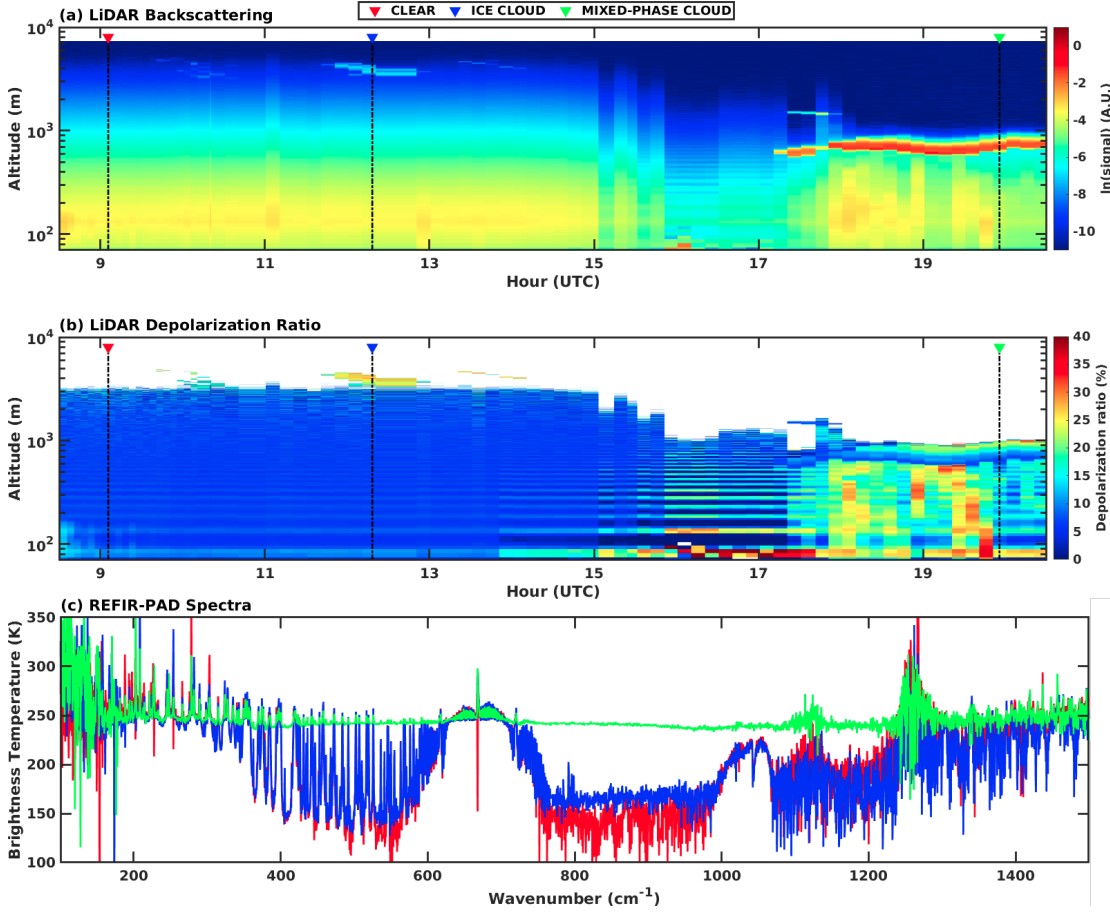

**Figure 2.** (a) LiDAR backscattering and (b) depolarization ratio for 2013, $2^{nd}$ January. (a) and (b): Different sky conditions are highlighted in correspondence of vertical dashed line. A red triangle indicate clear sky, blue triangle is used for ice cloud, and green triangle for mixed-phase cloud. (c): REFIR-PAD spectra in correspondence of the three sky conditions highlighted in the upper and middle panels. The same color code is used.

Since 2005, the Concordia station has provided hourly measurements of air temperature, pressure at the surface level, relative humidity, wind speed, and wind direction. The snow temperature is measured at different depths from 5 cm to 10 m. These measurements began in December 2012. Radiosondes Vaisala RS92 are routinely released every day at 12 UTC, since 2006. They reach an altitude of about 18 km in wintertime and about 25 km in the summer. All these data are made available by the Antarctic Meteo-Climatological Observatory and a subset of them is used in this study.





## 3 Cloud Identification and Classification Algorithm


The Cloud Identification and Classification (CIC) is a machine learning algorithm, based on the principal component analysis (PCA), able to classify an input spectrum ($\mathcal{L}$) as representative of a specific class, characterized by the elements contained in multiple groups of spectra used as training sets (TSs). The algorithm is based on the analysis of the measured spectra only and does not require any ancillary information or forecast model output data for the classification. The classification accounts

for the spectral features of the observed brightness temperature (BT), compared to the characterizing spectral features of each training set. A brief description of the algorithm is provided below; more details can be found in Maestri et al. (2019b).

For each class $X$ (i.e., clear sky, ice cloud, and mixed-phase) a set of spectra is used to set up a training set defining the variability within the class:

$$TS_X = TS_X(\tilde{\nu}, j) \tag{1}$$

where $\tilde{\nu}$ is the wavenumber, and $j = 1, \ldots, J$ refers to the $j^{th}$ element (spectrum) of the TS. The information content of the TSs is evaluated by computing the eigenvalues ($\lambda$) and the eigenvectors ($\epsilon TS$) of the TS covariance matrices:

$$[\lambda_X, \epsilon TS_X] = \mathrm{eig}(\mathrm{cov}(TS_X)) \tag{2}$$

The procedure also accounts for a spectral noise removal operation. This is performed by accounting only for a limited number of principal components, defined by Turner et al. (2006) as the first $P_0$ eigenvalues, out of P total components, that

minimize the *indicator function*:

$$IND(p) = \frac{RE(p)}{(P - p)^2} \tag{3}$$

where $p = 1, \ldots, P - 1$ refers to the $p^{th}$ principal component and the *real error RE* is defined as:

$$RE(p) = \sqrt{\frac{\sum_{i=p+1}^{P} \lambda_{X,i}}{J(P - p)}} \tag{4}$$

Each input spectrum $\mathcal{L}$ is then analysed by defining the extended training sets (ETS), that are the original TSs plus the input

spectrum itself:

$$ETS_X = [TS_X(\tilde{\nu}, j), \mathcal{L}(\tilde{\nu})] \tag{5}$$

and by computing the eigenvectors ($\epsilon ETS$) of each ETS covariance matrix.





The classification is performed through a parameter called similarity index (SI) that evaluates the variation of the information content in the ETS with respect to the original TS (for each class):

$$SI_X = 1 - \frac{1}{2P_0} \sum_{p=1}^{P_0} \sum_{\tilde{\nu}} |\epsilon ETS_X(\tilde{\nu},p)^2 - \epsilon TS_X(\tilde{\nu},p)^2| \tag{6}$$

The SI is a normalized index, where a value close to 1 means high similarity, and a value close to 0 means low similarity. As an example, if the input spectrum is measured in clear sky, the information content of the $ETS_{CLEAR}$ would be similar to the original $TS_{CLEAR}$, and their eigenvectors will also be very similar, due to the low additional information content from the input spectrum.

For this study, as previously indicated, three classes are defined: clear sky, ice clouds and mixed-phase clouds. Consequently, three training sets are arranged, each one containing spectra representative of that particular class. For each observation the operation described in Equation 6 is performed for 2 classes at a time. In our case, 3 SIs are obtained derived from the mutual comparison of the 3 classes. From these, a vector of similarity index differences (SID) is defined:

$$SID(1) = SI_{CLEAR} - SI_{ICE\ CLOUD}$$
$$SID(2) = SI_{CLEAR} - SI_{MIXED-PHASE\ CLOUD}$$
$$SID(3) = SI_{ICE\ CLOUD} - SI_{MIXED-PHASE\ CLOUD} \tag{7}$$

The classification of the input spectrum is performed in accordance with the logical diagram of Figure 3. The diagram shows the comparison between specific couples of SI (yellow boxes). The partial results of each comparison are represented by white boxes. If one class prevails over the other two, a classification is reached and the final output is provided (green boxes in the Figure).

The comparison between the SI of the classes is called *elementary approach*. This methodology is based on a very sim-
ple classificator, the SID, which works properly when each class is characterized by specific spectral features that make the elements of the class easily distinguishable from those pertaining to other classes. This is clearly very difficult to attain for some classes such as, for example, the clear sky class and the cirrus cloud class. The classification of clouds over the Antarctic Plateau is particularly challenging, primarily because of the generally small cloud optical depths whose IR spectral character-istics are very similar to that of the clear sky. The selection of the spectra contained in each training set is crucial, as it is in
every classification algorithm. In fact, the selected elements must represent the entire class characteristics and variability to perform a correct classification.

Maestri et al. (2019b) suggested that better results can be obtained when a classificator optimization is performed a-priori by using a methodology called *distributional approach*. When applied to a set of observations, a perfect classifier would ideally generate a bimodal SID distribution for each comparison between two classes, splitting the elements in two separate groups.
This class separation is difficult to achieve in reality and the amount that elements overlap depends on many factors, including the spectra used to define the training sets. To mitigate the issue, the CIC is applied separately to each training set element.



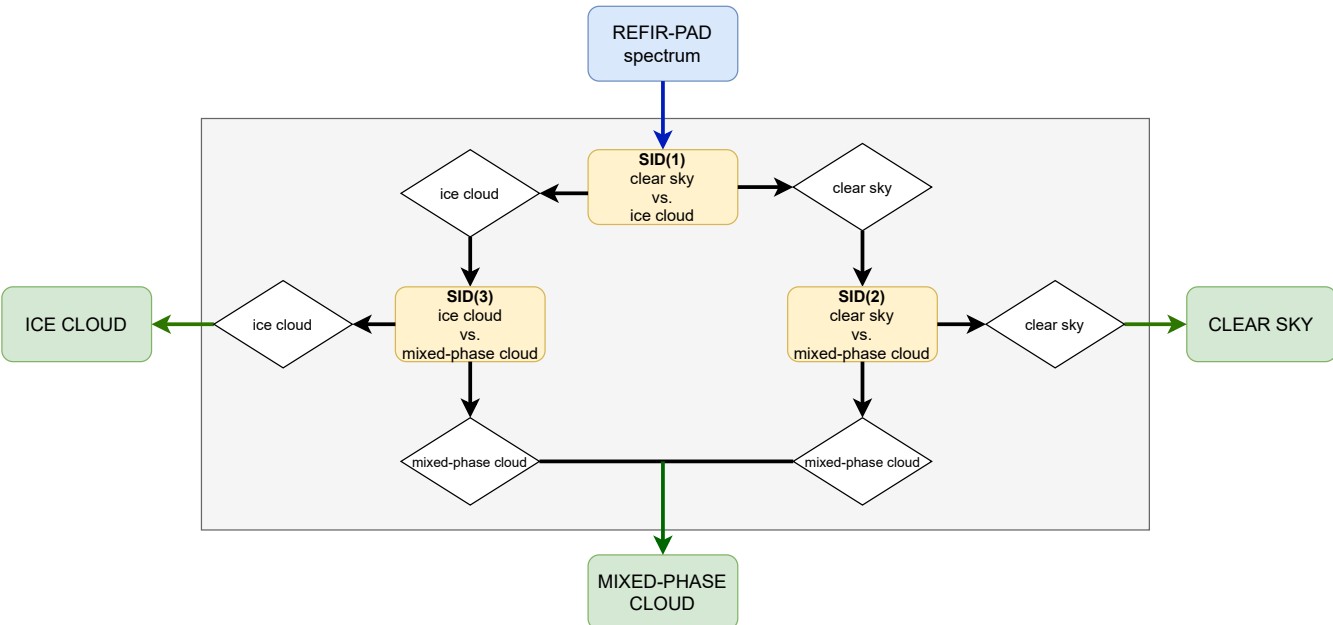

**Figure 3.** Logical diagram of the classification process performed by the CIC algorithm for the definition of the clear sky, ice cloud and mixed-phase cloud classes.

Based on the result for each spectrum of known class, an evaluation can be made for the SID distribution for the entire set of each class. Through this analysis of the SID distributions, an optimal SID delimiter can be defined to maximize the correct classification of the training set elements. The delimiters, that can be different from zero, are set according to the classification

results to optimize the algorithm performance. An example of the SID distribution based on the training set spectra and of the elementary and the distributional approaches is provided in Figure 4. The CIC is applied to the training set spectra of clear sky and mixed-phase clouds. The elementary method (left panel) classifies as clear sky (blue shaded area) all the spectra with SID$\leq$0 and as mixed-phase cloud (red shaded area) all the spectra with SID>0. This methodology misclassifies some of the mixed-phase cloud training set spectra (red histogram). The distributional method (right panel) maximises the classification

performance by defining a new delimiter between clear and mixed-phase cloudy scenes. In this example, the new delimiter is set at SID=$-0.15$, so that most of the TS spectra are correctly classified. See Maestri et al. (2019b) for a description of the computation of the delimiter. Once the delimiters (DEL) are defined for each class couplet, the classification is performed by using a Corrected Similarity Index Difference (CSID):

$$CSID(1) = SI_{CLEAR} - SI_{ICE\ CLOUD} + DEL_{(clear\ -\ ice\ cloud)}$$
$$CSID(2) = SI_{CLEAR} - SI_{MIXED-PHASE\ CLOUD} + DEL_{(clear\ -\ mixed-phase\ cloud)}$$
$$CSID(3) = SI_{ICE\ CLOUD} - SI_{MIXED-PHASE\ CLOUD} + DEL_{(ice\ cloud\ -\ mixed-phase\ cloud)} \tag{8}$$





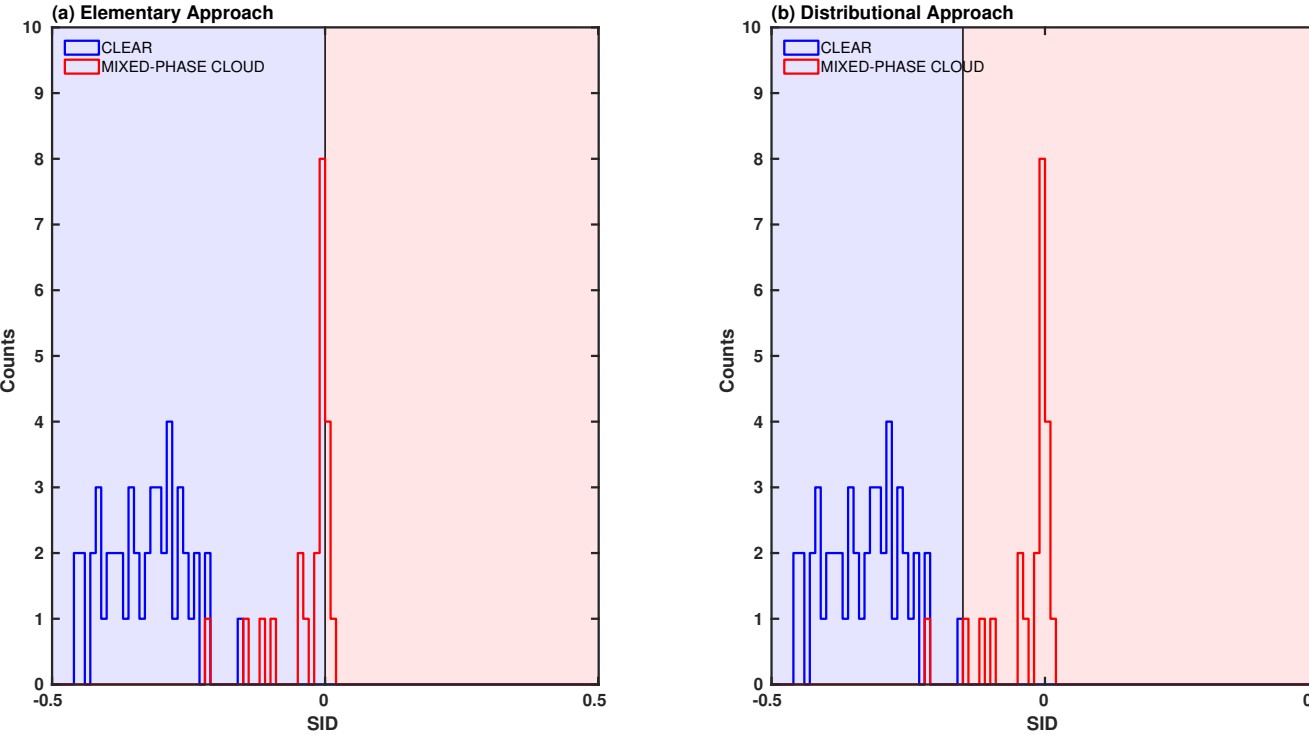

**Figure 4.** Example of (a) the CIC elementary approach and (b) the distributional approach applied to the training set elements of clear sky (49 spectra) and mixed-phase clouds (22 spectra), in the warm season (November-March). The clear sky (blue histogram) and the mixed-phase cloud (red histogram) training set elements are classified according to the SID as clear sky (blue shaded area) or mixed-phase cloud (red shaded area) scenes. Only 76% of the spectra are correctly classified using the elementary approach, while 99% of the spectra are correctly classified using the distributional method.

The entire classification procedure, schematically described in Figure 3, is then performed by the new classifier CSID in place of the SID. Due to the better performance, the distributional method is preferred and applied in this study.

## 3.1 Training and Test Sets

Spectra used to populate the training sets are chosen from a set of manually classified observations. The identification is performed by visually inspecting the co-located LiDAR backscatter and depolarization profiles in accordance with the criteria described in Section 2. Each training set contains a limited number of spectra from the REFIR-PAD database, aiming at describing the variability of atmospheric conditions over the Concordia station. Due to the intense variations of the environmental conditions, the training sets are defined for two macro-seasons: a warm season (November-March) and a cold season (April-October). The choice is also supported by the fact that mixed-phase clouds are extremely rare in the cold macro-season. Ricaud et al. (2020) observed the occurrence of supercooled liquid water clouds during the warm macro-season only, with the largest frequency occurring in December and January. Listowski et al. (2019) also observed that the fraction of supercooled liquid





**Table 2.** Number of spectra used in each TS, according with the macro-season.

| Season / class | clear sky | ice cloud | mixed-phase cloud |
|---|---|---|---|
| November-March | 49 | 30 | 22 |
| April-October | 64 | 37 | – |

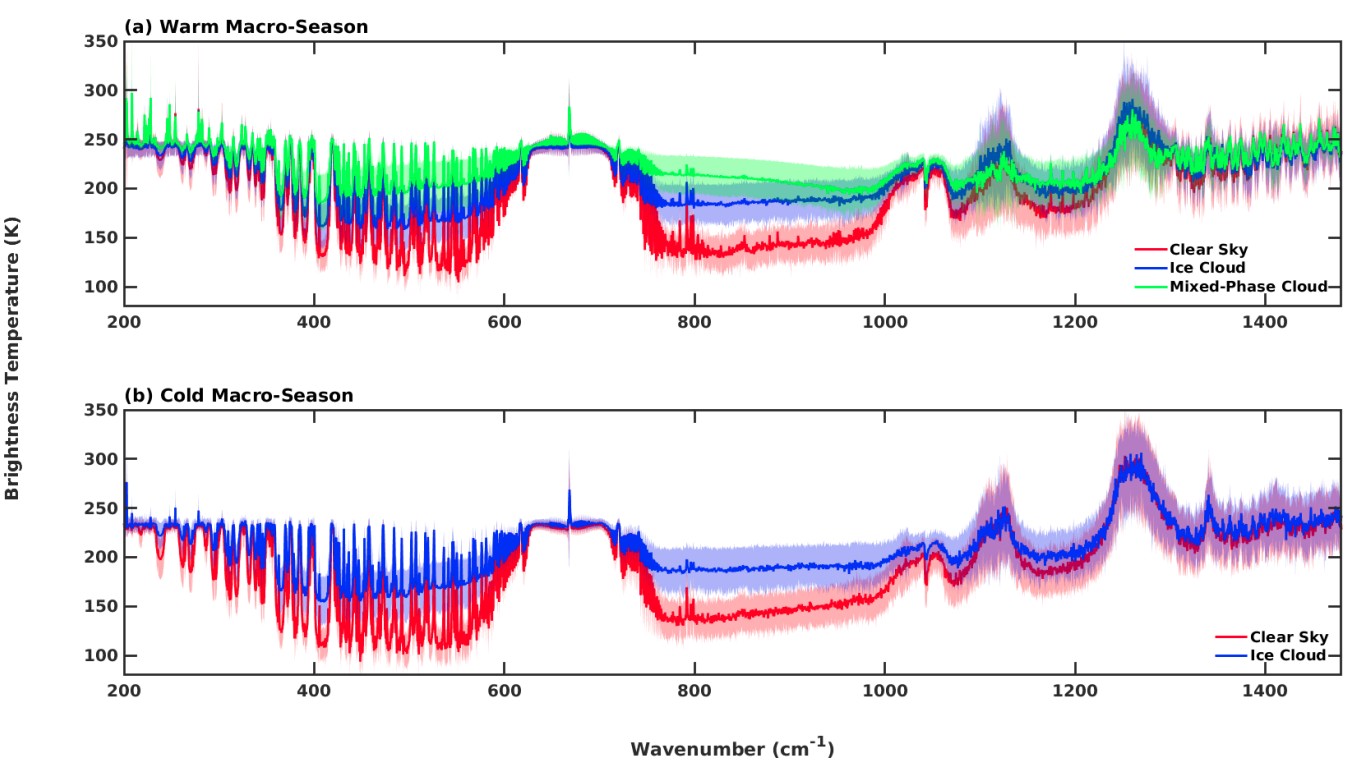

**Figure 5.** Average BT (solid line) ±1 standard deviation (shaded area) for TS elements of the (a) Warm and (b) Cold macro-seasons.

water-containing clouds in the Antarctic Plateau varies between 10%, in the summertime, and 0%, in winter. Therefore, three training sets for the warm macro-season are defined: clear sky, ice cloud, and mixed-phase cloud. For the cold macro-season, only the clear sky and ice cloud training sets are used. Table 2 summarizes the number of spectra for each TS and macro-season.

Mean spectra in terms of BT (solid lines) and their standard deviations (shaded area) are presented in Figure 5 for the training sets used for both macro-seasons. Differences between the mean spectra of the different classes are observed in the window channels located between 400 and 600 cm$^{-1}$, and 800 and 1000 cm$^{-1}$. Note that in IR window regions (transparent channels) the standard deviation of the clear sky spectra is usually lower than that of the cloudy spectra, which account for a wider signal variability in these bands. Furthermore, the clear sky signal is very low at window wavenumbers, that results in a very low signal to noise ratio.





Once the TSs are defined, the DELs are computed (as described in Section 3) and the CIC is ready to ingest the REFIR-PAD spectra and provide their classification. To evaluate the CIC performance and optimize its set-up, a test set is analysed of 1726 pre-classified spectra collected in 2013. The test set is composed of 559 clear sky, 1022 ice cloud, and 145 mixed-phase cloud spectra. These spectra were previously classified by using the co-located LiDAR backscatter and depolarization profiles. An example is provided in Figure 2. We define the sky condition as that observed when the REFIR-PAD starts its measurement.

Then, the visually classified REFIR-PAD spectra are associated to the sky conditions encountered at the beginning of each measurement.

### 3.2 CIC performance and optimization

The CIC algorithm is applied to the test set spectra by accounting for their BT in different spectral intervals. This operation is performed to find the optimal spectral interval that maximizes the classification results for each class (clear sky, ice cloud,

mixed-phase cloud). Multiple runs of the CIC algorithm are performed on the same test set, by applying it to different spectral ranges. Specifically, the starting wavenumber is moved, at steps of 20 cm$^{-1}$, in the $200 - 600 cm^{-1}$ band, and the ending wavenumber is moved between 960 and 1480 cm$^{-1}$.

  The algorithm performance during this process is assessed by evaluating the Threat Score (ThS). A confusion matrix is used to compute the ThS of each class and for each considered spectral interval. Each individual spectrum can be classified correctly

as a member of its class (i.e class $A$), or incorrectly as a member of a different class (i.e. class $B$ or $C$). With this symbolism, the spectrum classification is interpreted in terms of:

  – True positive (TP): the spectrum belong to class $A$ and it is properly classified in class $A$.

  – True negative (TN): the spectrum does not belong to class $A$ and it is properly classified in its class of pertinence ($B$ or $C$).

– False positive (FP): the spectrum belongs to class $B$ or $C$ but it is misclassified in class $A$.

  – False negative (FN): the spectrum belongs to class $A$ but it is misclassified in class $B$ or $C$.

  Given the above possibilities, for each class the threat score is defined as:

$$ThS = \frac{TP}{TP + FN + FP} \tag{9}$$

that accounts for the correctly classified spectra (TP) in the class and penalizes all the misclassified occurrences (FN and FP).

A ThS value of 1 means that there are no misclassified spectra.

  Based on the results obtained for each of the combinations of starting and ending wavenumbers, the ThS is calculated for each class (clear sky, ice cloud, mixed-phase cloud). The weighted mean ThS values, that account for the total number of cases in each class, are also calculated. In the upper left panel (a) of Figure 6 the mean ThSs are plotted as a function of the starting and ending wavenumbers. The other panels in this figure (b-c-d) show results for the three specific classes. The ThS values span

from 0.487 to 0.966 in accordance with the selected interval and the given class. For intervals ending with wavenumbers larger



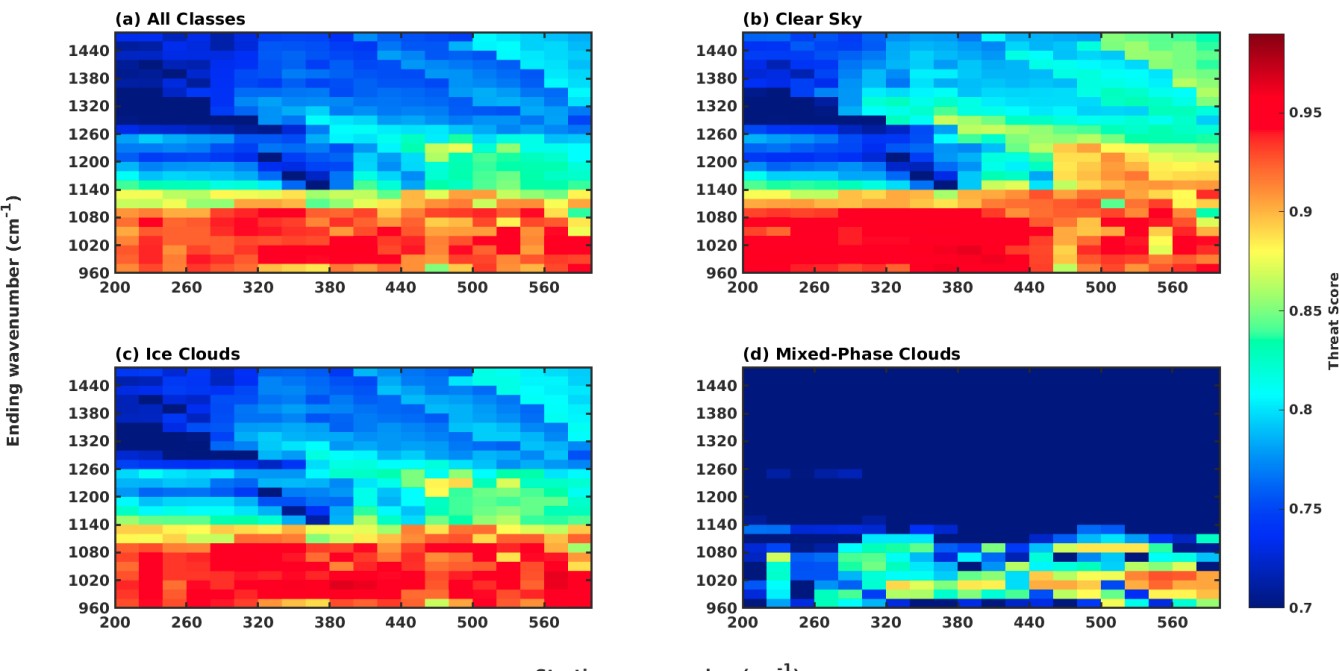

**Figure 6.** Threat Scores for the test set as a function of different spectral intervals ingested by the CIC algorithm. The (a) All Classes threat score is a weighted mean of the threat scores computed for each class: (b) clear sky, (c) ice cloud, and (d) mixed-phase cloud.

than 1140 cm$^{-1}$, the ThS decreases considerably for all the classes. This is likely associated to the noise of the REFIR-PAD sensor which increases considerably above 1200 cm$^{-1}$ and degrades the classification results. When the ending wavenumber is set to values between 980 and 1080 cm$^{-1}$, the ThS is very high (larger than 0.9) for all the starting wavenumbers below 400 cm$^{-1}$, both for clear sky and ice clouds. The spectral interval 380–1000 cm$^{-1}$ performs the best for classification of both clear

sky and ice clouds, where the ThS values are 0.963 and 0.966, respectively. The classification of mixed-phase clouds is slightly less robust compared to the other two classes, and the best spectral interval is 540–1020 cm$^{-1}$ with a ThS of 0.927.

When accounting for all the classes, the most performing spectral range for clear and cloud identification and classification is the 380–1000 cm$^{-1}$ interval. The result is dependent on sensor characteristics and for this study it is specifically driven by the REFIR-PAD spectral resolution and noise features. The optimal interval for the classification is also dependent on many other

parameters, among which are the type and number of classes considered, the observation geometry (e.g. satellite or ground based), the observing location, and the mean atmospheric conditions. Because the water vapor content is extremely low, the ground-based measurements on the Antarctic Plateau allow the full exploitation of the FIR spectral range. These channels would be totally opaque for upward observations in regions of increased water vapor content such as the tropics. The selected spectral range (380–1000 cm$^{-1}$) highlights the fundamental role of the FIR part of the spectrum in the cloud identification and

classification.



The results of the CIC classification applied to the test set using the 380–1000 cm$^{-1}$ are summarized in Table 3. The Table reports the number of spectra per class in the test set, the CIC Hit Rates (HR), in percentage, and the threat scores. The HR for a class (i.e. $A$) is defined as:

$$HR_A = \frac{N_A^{\mathrm{CIC}}}{N_A^{\mathrm{true}}} = \frac{TP}{TP + FN} \tag{10}$$

where $N_A^{\mathrm{CIC}}$ is the number of occurrences of the class $A$ that are correctly identified by the CIC (corresponding to the TP in the confusion matrix). $N_A^{\mathrm{true}}$ is the total number of elements in class $A$ of the dataset, and corresponds to TP+FN of the class $A$.

The overall performance is that almost 98% of spectra are correctly classified. Only a small percentage (less than 1%) of cloudy spectra (ice clouds plus mixed-phase clouds) is misclassified as clear sky, and about 2% of the clear sky spectra are
erroneously identified as ice clouds. Note that in case of mixed-phase clouds the CIC is able to identify the presence of the cloud in 99.3% of the cases even if for the 8.3% the cloud phase is classified as ice instead of mixed-phase. This is actually a very reasonable performance considering that, as noted before, most of the mixed-phase clouds are composed of a layer of super-cooled liquid phase near the cloud base and ice phase particles close to the cloud top. For optically thin clouds, the definition of the phase can be problematic.

Sensitivity studies on the identification of mixed-phase clouds are performed assuming a cloud layer of total optical depth at 900 cm$^{-1}$ of 2 in which the base layer is composed of liquid water and the upper layer is occupied by ice particles. Results (not shown here) demonstrate that for the bottom layer of liquid phase with OD larger than 0.1-0.3 the cloud is identified as mixed-phase, otherwise, it is classified as ice cloud. This demonstrates that the algorithm is very sensitive to the presence of thin liquid water layers at cloud base. Nevertheless, it is also possible to incur in situations in which a very thin layer of liquid
water is at the base of a thicker ice layer and the spectral signal measured at the ground is interpreted by the CIC algorithm as exiting from an ice cloud.

### 3.2.1    Test Set Misclassified Spectra

Each of the misclassified cases is inspected visually to understand the main causes of error in the CIC classification. It appears that the misclassification of clear sky as ice cloud, and vice-versa, occurs primarily for spectra taken during the cold macro-
season. The misclassification in this case is associated with the: (a) presence of a very thin cirrus cloud; (b) REFIR-PAD measurements taken over a period of time in which the observed scene is changing (i.e. the measuring time encompasses both clear sky and cloudy sky); (c) presence of suspended particles near the surface (e.g., diamond dust, wind-blown snow, or combustion products produced by the generator that heats Concordia Station).

During the warm macro-season, a small percentage of mixed-phase clouds are misclassified as either clear sky or ice clouds.
In some cases, ice clouds are misclassified as mixed-phase clouds; this happens mostly when the ice cloud spectra are characterized by a high BT in the main window region.



**Table 3.** Test set classification performed by CIC, using the optimal spectral range 380–1000 cm$^{-1}$.

| Class | # Spectra | Hit Rate | Threat Score |
|---|---|---|---|
| **Clear Sky** | **559** | **98.0% - Clear Sky** <br> 2.0 % - Ice Cloud <br> 0.0 % - Mixed-Phase Cloud | **0.963** |
| **Ice Cloud** | **1022** | 0.9 % - Clear Sky <br> **98.7 % - Ice Cloud** <br> 0.4 % - Mixed-Phase Cloud | **0.966** |
| **Mixed-Phase Cloud** | **145** | 0.7 % - Clear Sky <br> 8.3 % - Ice Cloud <br> **91.0 % - Mixed-Phase Cloud** | **0.886** |
| **Total** | **1726** | **97.9 % - Correct** <br> 2.1 % - Misclassified | **Weighted mean:** <br> **0.958** |

## 4 Results

The 380–1000 cm$^{-1}$ spectral interval is used to run the CIC algorithm over the entire REFIR-PAD dataset, comprising measurements from year 2012 through 2015. In Figure 7, the CIC classifications are compared with co-located LiDAR depolarization data for two different days. For each REFIR-PAD observation, the classification is reported as a colored triangle in the upper part of each panel. As previously discussed, low values of LiDAR depolarization together with large values of the backscattering signal (not shown) indicate the presence of liquid water phase in the cloud layer, while high depolarization values are observed in presence of ice clouds. The upper panel of Figure 7 shows the presence of a mixed-phase cloud over the Concordia station from about 10:00 UTC until the nighttime of the 3$^{rd}$ of January 2014. The presence of the cloud and its thermodynamic phase are correctly identified and classified by the algorithm. Between hours 21:00 and 22:30 UTC, CIC identifies a spectral signal characteristic of ice clouds that corresponds to larger values of the depolarization ratio measured by the LiDAR. On the 1$^{st}$ of August 2014 (lower panel of Figure 7), the LiDAR depolarization shows that the day starts with a precipitating ice cloud, followed by clear sky conditions from 15:00 UTC. For this case, both the clear sky and the ice cloud are correctly detected by the CIC algorithm.

The results of applying the CIC to the full available REFIR-PAD dataset are provided in terms of percentages, defining the occurrence of each class with respect to the total number of analysed spectra. An error can be associated to the percentage occurrence, exploiting the HRs derived in the analysis of the test set. With the use of Equation 10 for the HR definition for the class $A$:

$$N_A^{\text{CIC}} = N_A^{\text{true}} \times HR_A \tag{11}$$



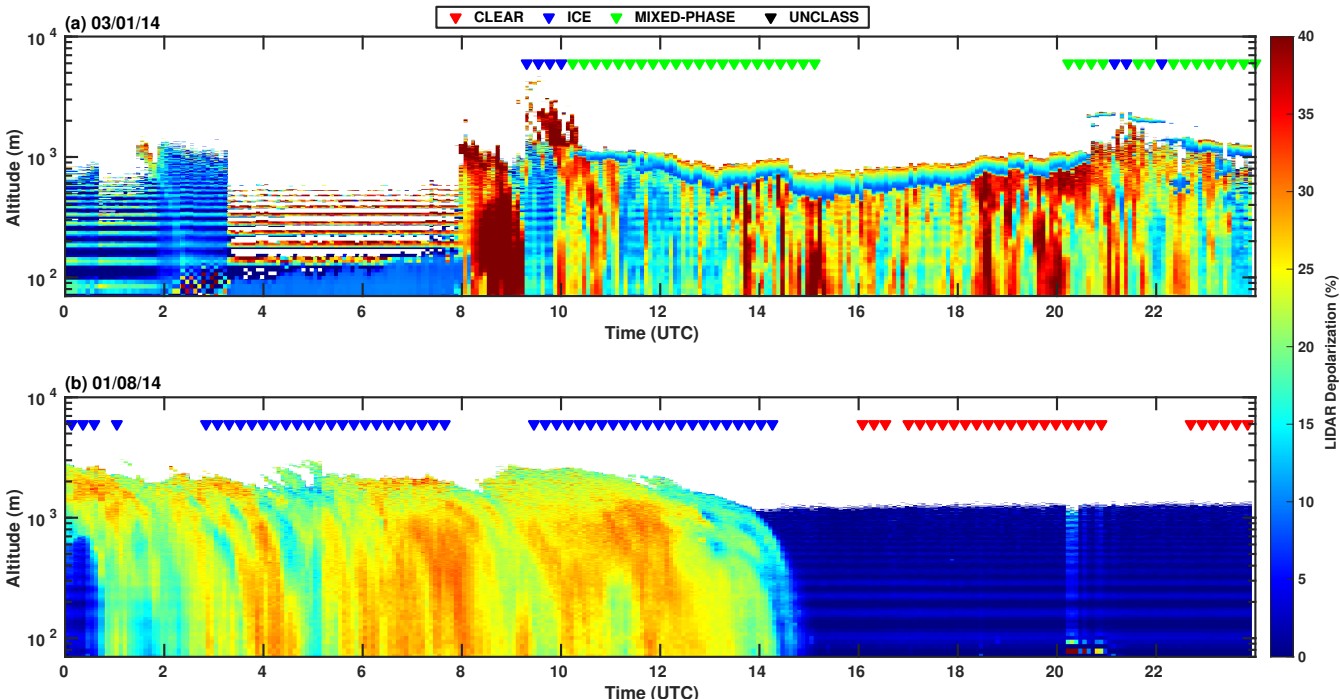

**Figure 7.** LiDAR depolarization ratio on (a) $3^{rd}$ of January 2014 and (b) $1^{st}$ of August 2014. The triangles mark the REFIR-PAD observations. The color code indicates the CIC classification: red for clear sky, blue for ice cloud, green for mixed-phase cloud, and black for unclassified spectra.

the number of misclassified spectra ($N_A^{\mathrm{err}}$) of class A can be written as:

$$N_A^{\mathrm{err}} = N_A^{\mathrm{true}} \times (1 - HR_A) \tag{12}$$

Through combination of Equation 11 and Equation 12, it is possible to remove the term $N_A^{\mathrm{true}}$ which is unknown for results applied to the entire dataset. The following relation is then derived:

$$N_A^{\mathrm{err}} = N_A^{\mathrm{CIC}} \times \frac{(1 - HR_A)}{HR_A} = N_A^{\mathrm{CIC}} \times \left( \frac{1}{HR_A} - 1 \right) \tag{13}$$

The relative error ($\epsilon$), associated to the classification of the elements of class $A$, is obtained by dividing the number of misclassified $A$ spectra to the total number of spectra $N_{A+B+C} = N_{\mathrm{TOT}}$:

$$\epsilon_A = \frac{N_A^{\mathrm{err}}}{N_{\mathrm{TOT}}} = \frac{N_A^{\mathrm{CIC}}}{N_{\mathrm{TOT}}} \times \left( \frac{1}{HR_A} - 1 \right) \tag{14}$$

Note that the HR values associated with the individual classes for the entire dataset are unknowns. However, it is assumed that the CIC scores over the test set spectra are representatives of the performances that are obtained over the full dataset.





Therefore, the HRs obtained for the test set analysis (see back Table 3) are used in place of the dataset HR in Equation 14. Thus, for the class $A$, the percentage classification error is simply:

$$\epsilon_A\% = \frac{N_A^{\mathrm{CIC}}}{N_{\mathrm{TOT}}} \times \left( \frac{1}{HR_A} - 1 \right) \times 100 \tag{15}$$

where $N_A^{\mathrm{CIC}}$ is the number of spectra identified by CIC as member of class $A$ and $N_{\mathrm{TOT}}$ is the total number of spectra in the entire dataset. The $HR_A$ is obtained from the application of CIC to the test set and is thus known a-priori. Note that for a

small number of false positives ($FP \ll TP$) the HR for class $A$ is very similar to the ThS for the same class. CIC provides very small values of FP when applied to the test set, with respect to TP values: 2% for ice clouds and clear sky, and about 3% for mixed-phase clouds.

### 4.1 Sky classification: 4 years averages and inter-annual variability

A total of 87960 REFIR-PAD spectra are analysed from the dataset spanning over the time range 2012–2015. From this set,

only 202 spectra are used as TS, and the other 87758, which include the test set elements, are ingested by the CIC algorithm to evaluate the cloud occurrence over the Concordia station. The classification results are shown in Table 4 as percentages for clear sky, ice clouds, mixed-phase clouds, and unclassified spectra. The entire dataset and individual years classifications are presented, as well as the estimated percentage uncertainties (see Equation 15). On average, the clear sky is detected in almost 72% of the cases, with ice cloud occurrence of about 25% and mixed-phase cloud occurrence of less than 3%. The inter-annual

variability of total cloud occurrence in the Antarctic Plateau, the sum of ice and mixed-phase clouds, spans between about 23 and 31%. This percentage interval is in accordance with the observations from Adhikari et al. (2012), who analysed data from CloudSat and CALIPSO between 2006 and 2010 and reported percentages spanning between 20–30% interval. From our analysis the cloudiest year in the 2012–2015 period is 2012, with a value of 31.2%. This is almost identical to what observed in 2015 (cloud occurrence is 31.1% in this case), with the difference being that in 2012 there was a significantly larger fraction

of mixed-phase clouds than in 2015 (5.8% and 1.5% respectively).

Mean temperatures at the surface level for the entire dataset and for each single year are also reported in Table 4. Temperatures are measured every hour at the Concordia station and are linearly interpolated in time to be associated with the REFIR-PAD measurements and the corresponding CIC classifications. The last row of Table 4 provides information only for the months of the warm macro-season from November to March. The results suggest a positive correlation between mean air

temperatures at surface level in the Warm macro-season and the occurrence of mixed-phase clouds. Note that mixed-phase clouds are present only for months from November to March. The temperature and mixed-phase cloud correlation could indicate that warm temperatures are favorable for mixed-phase clouds formation or that the presence of warm liquid clouds implies a stronger cloud forcing at the surface and, consequently, an increase in the temperature values near the ground. Ice clouds are observed during the entire year. In contrast with mixed-phase clouds, their occurrence does not seem correlated to the mean

air temperature at the surface. Note that the maximum occurrence of ice clouds is observed during year 2015, which had the lowest mean value of surface air temperature in the 4 years time range.





**Table 4.** CIC classification results for the whole REFIR-PAD spectra dataset (2012–2015) and for single years. The associated uncertainties are computed using Equation 15. Mean air temperatures at surface level for the entire period and for each year are also reported. The last row refers to mean air temperatures at surface level computed for the months from November to March (Warm season).

| CIC CLASSIFICATION | ENTIRE DATASET (%) | 2012 (%) | 2013 (%) | 2014 (%) | 2015 (%) |
|---|---|---|---|---|---|
| **CLEAR SKY** | **72.3 ± 1.5** | 68.6 ± 1.4 | 75.1 ± 1.5 | 76.3 ± 1.5 | 68.8 ± 1.4 |
| **ICE CLOUD** | **24.9 ± 0.3** | 25.4 ± 0.3 | 22.8 ± 0.3 | 21.1 ± 0.3 | 29.6 ± 0.4 |
| **MIXED-PHASE CLOUD** | **2.7 ± 0.3** | 5.8 ± 0.6 | 2.0 ± 0.2 | 2.5 ± 0.2 | 1.5 ± 0.2 |
| **UNCLASSIFIED** | **0.1 ± 0.1** | 0.2 ± 0.1 | 0.1 ± 0.1 | 0.1 ± 0.1 | 0.1 ± 0.1 |
| **Mean T (°C)** | -53.5 | -49.6 | -54.5 | -53.4 | -55.0 |
| **Warm season Mean T (°C)** | -40.2 | -37.6 | -41.0 | -40.7 | -41.1 |

## 4.2 Seasonal clear sky and cloud occurrence

Seasonal averages of cloud occurrence are computed for the entire dataset and presented in Table 5. The Table also reports the number of spectra observed in each season, which show that the data are homogeneously distributed over the course of the year, and the mean air temperatures. The mean total cloud occurrence varies from the minimum value of 23.9% detected in spring (SON) to the maximum value of 33.2% in the cold winter season (JJA). The dominant cloud occurrence and thermodynamic phase is ice. During the austral summer, the occurrence of ice clouds is the smallest. However, for the same season, the occurrence of mixed-phase clouds reaches its maximum over Concordia Station (10.9%). It is interesting that during summer, more than one third of the clouds over Concordia is of the mixed-phase type. The occurrence of mixed-phase clouds in summer is in line with the analysis performed by Listowski et al. (2019), who analysed DARDAR data (Delanoë and Hogan, 2010; Ceccaldi et al., 2013) based on combined observations from CloudSat and CALIPSO satellites, in the period 2007–2010. The same authors, by performing a visual analysis of the geographical distribution of the clouds containing liquid water particles, estimate that during the other seasons (MAM, JJA, and SON), the occurrence of mixed-phase clouds is close to 0%, in the region around the Concordia station.

Seasonal occurrences for each class are analysed in combination with meteorological parameters encountered during the corresponding REFIR-PAD measurements. In Figure 8, the percentage distribution of each class seasonal occurrence is reported as a function of the air surface temperature, with histograms binning of 7°C. The same color code is adopted here that was used previously: clear sky in red, ice clouds in blue, and mixed-phase clouds in green. The number of REFIR-PAD measurements for each bin is reported at the base of the histograms. Over the four years, the surface air temperature (in correspondence of REFIR-PAD measurements) varies between a minimum of $-81.3$°C and a maximum of $-15.8$°C. With the exception of the spring season (SON, lower right panel of Figure 8), the results show that the detected cloudy sky occurrence increases (clear skies decrease) as surface air temperature increases. This holds both for ice and mixed-phase clouds. In the winter season (JJA, lower left panel of Figure 8), for surface air temperature larger than $-43.3$°C the CIC identifies only ice cloud conditions. Note





**Table 5.** Mean seasonal occurrences of clear sky, ice clouds, and mixed-phase clouds at Concordia station. Mean surface air temperatures are reported for each season.

|  | DJF | MAM | JJA | SON |
|---|---|---|---|---|
| # spectra | 21209 | 21093 | 22395 | 23061 |
| **CIC CLASSIFICATION** | (%) | (%) | (%) | (%) |
| **CLEAR SKY** | 71.1 | 75.1 | 66.8 | 76.1 |
| **ICE CLOUD** | 17.6 | 24.7 | 33.2 | 23.8 |
| **MIXED-PHASE CLOUD** | 10.9 | 0.2 | 0.0 | 0.1 |
| **UNCLASSIFIED** | 0.4 | 0.0 | 0.0 | 0.0 |
| **Mean T (°C)** | -34.9 | -61.0 | -65.0 | -52.2 |

that the winter and spring seasons have the largest variation in the air surface temperatures. In the winter season, extremely

low temperatures (below $-70°C$) are very frequent and result from the lack of insolation, the dry atmospheric conditions, and the absence of clouds. In the same season, higher surface temperatures are measured mainly in correspondence of cloudy sky conditions. The downwelling longwave radiation from cloud layers contributes to the surface radiative forcing and mitigates the temperature of the cold season. Over the four-year period the average winter surface temperature in clear sky conditions is $-67.9°C$, while in presence of ice clouds is $-59°C$.

A similar analysis is performed by relating clear and cloudy sky occurrences to measurements of surface relative humidity and surface pressure. Results (not shown here) indicate that the highest values of relative humidity tend to occur with the highest percentage of clouds for all the seasons except spring. The highest mean values of surface pressure in the summer season tend to occur with the highest percentages of mixed-phase clouds (not shown). Unclassified spectra are obtained only in the summer season and in correspondence of very high values of surface pressure, air temperature, and relative humidity.

Surface wind measurements are also analysed and related to CIC classification results for each season. The values of wind speed and direction closest in time to the REFIR-PAD measurements are used. Wind roses are built considering the bias correction methodology proposed by Droppo and Napier (2008), which indicates the necessity of weighting the contribution of each direction to correctly represent them in the wind roses.

In Figure 9, the wind roses for each season and class are shown. Clear sky cases correspond to about 70% of all occurrences

in all seasons and are associated with a surface level wind that blows predominantly from South and South-West. Higher wind intensities are found in springtime. An additional wind component from the West is observed in summer, but is negligible in the other seasons. When ice clouds are present, the dominant surface wind direction is from the South-East, and the wind intensity is larger than in clear sky conditions on average (7.7 m/s versus 6.1 m/s). Note that non-negligible occurrences of surface wind from the North-East are observed only when mixed-phase clouds are detected, especially during the fall (MAM)

season. This component overlaps with the dominant South-East wind component found both in summer and autumn. The wind rose for mixed-phase clouds in the spring season (SON) is reported for completeness but is affected by the very few number of cases detected. Even if very preliminary, the analysis of the surface wind direction in presence of different sky conditions





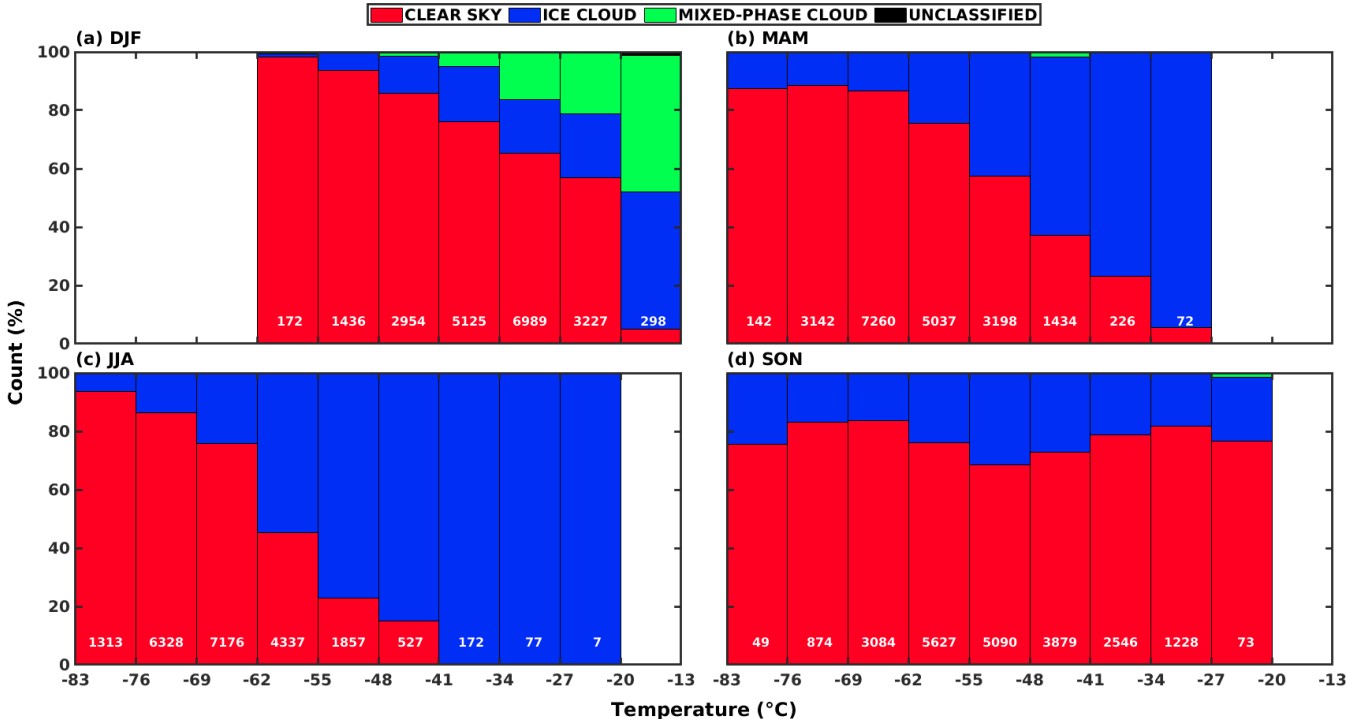

**Figure 8.** Histograms of the seasonal occurrence of the analysed sky conditions as a function of the surface air temperature. (a) DJF - summer, (b) MAM - autumn, (c) JJA - winter, and (d) SON - spring. The number of observations for each 7°C bin is reported at the base of each histogram.

highlights some correlations between the wind component and the clear sky or cloud occurrence. Winds from the South and West directions at the Concordia station are from the Antarctic Plateau, where the drier air is supposedly found. Winds from
the South-East and East directions suggest a moisture transport from the Ross sea to the region of the Concordia station. The correlations are far from being conclusive since the upper level winds and the back trajectories of the air masses have not been analyzed yet.

### 4.3 Monthly mean cloud occurrence: comparison with satellite data

CIC monthly mean cloud percentages (including ice and mixed-phase) for the period 2012–2015 are shown in Figure 10. The
black curve corresponds to the 4-years monthly average cloud occurrence, and the shaded grey area indicates the minimum and maximum CIC monthly values. The lowest average value is found in November (17%), while higher occurrences are observed during the winter months. The peak is located in August, with an average value of 39%. For the same month, the inter-annual variability is quite large as indicated by the extent of the grey area. As examples, in August the monthly mean values span from 31% to 62%, which is the highest derived occurrence, and in November from 1% (lowest registered value) to 37%.





**Figure 9.** Wind roses at Concordia station, for the four seasons (DJF, MAM, JJA, and SON) of the period 2012–2015. Clear sky, ice cloud, and mixed-phase cloud conditions are split into separate rows.

For comparison, the monthly mean cloud occurrences/fractions derived from level 3 (L3) satellite products are reported for the same period of time. According to WMO [1], the L3 satellite products are composed of variables mapped on uniform space-time grid scales, and are constructed to provide completeness and consistency for the anticipated users. These products type are frequently used to perform climate analysis and model evaluation (e.g. Stubenrauch et al., 2013; Webb et al., 2017). In practice, different data sets present specific strengths and limitations that are briefly described below.

The L3 products used in this work are derived from passive radiometric observations performed by the Moderate Resolution Imaging Spectroradiometer (MODIS) on board the TERRA and the AQUA satellite platforms, by the CALIOP on board the CALIPSO satellite, and by the CPR on board CloudSat satellite.

---

[1] World Meteorological Organization - http://www.wmo.int/pages/prog/sat/dataandproducts_en.php, last access: $21^{th}$ January 2021.





For MODIS L3 products, the occurrence by cloud type is not available, and the cloud fraction is used. This variable is computed as the ratio between the cloud covered pixels and the total number of pixels observed by both satellites platforms each month and is mapped in a global grid of 1° of latitude and longitude, which corresponds to an area of about 3000 km$^2$ in the region of the Concordia station.

Two types of MODIS L3 products are used in this study: MCD06COSP and MYD08/MOD08. The first one combines the observations from both AQUA and TERRA platforms (MCD06COSP_L3, MODIS Atmosphere Science Team, 2020). This product is based on a cloud mask which uses bands at visible and infrared wavelengths. Thus the cloud fraction information (solid blue in Figure 10) is not available in the Antarctic region during the dark season (between May and July). The second product is derived from each MODIS sensors on platforms separately (MYD08 for AQUA, and MOD08 for TERRA, MODIS Atmosphere Science Team, 2017). The MOD08/MYD08 L3 product is based on a cloud mask which exploits infrared bands when in absence of solar illumination. In this case, the monthly mean cloud fraction is available for all the seasons (dashed and dotted blue curves in Figure 10, for MODIS TERRA and AQUA L3 products, respectively).

In contrast to MODIS, the CALIOP and the CPR active sensors detect the cloud occurrence within vertical profiles. The L3 product from these sensors is a volume cloud occurrence, which considers the number of cloud observations along the vertical profiles that are mapped monthly on a regular grid. The CALIOP L3 product (CAL_LID_L3_Cloud_Occurrence-Standard-V1-00, Winker, 2018) is built on a grid map of 2.5° of longitude and 2.0° of latitude, which corresponds to an area of about 15000 km$^2$ in the region surrounding Concordia station. CALIOP mean occurrence is plotted in green in Figure 10. The L3 product from CloudSat (3S-RMCP, Haynes, 2019) is available in a grid of 5°x5° of latitude and longitude, that covers an area of about 75000 km$^2$ around the Concordia station. Cloudsat results are reported as a red curve in the Figure. From year 2011, the CPR on CloudSat collected data only in daylight hours due to a battery anomaly, so there is no record of cloud occurrence from CloudSat from April to August.

For each one of the MODIS, CALIOP and CPR sensors, the grid point that includes the Concordia station is used to retrieve the monthly L3 satellite product. Monthly time series of the cloud fractions, in the case of MODIS data, and of cloud occurrences, in case of CALIOP and CPR observations, are computed for the period 2012–2015. Results are compared with the cloud occurrence derived by the CIC algorithm over the Concordia station (Figure 10). Since the L3 products of the three sensors refers to multiple extent areas of observations (of the order of tens of thousands of km$^2$), some differences are expected not only between the ground-based observations analysed by CIC but also among the mean values of the L3 satellite products.

In months in which insolation is present, the lowest values are those derived from CALIOP products, in green in Figure 10. Despite the very low values, CALIOP is able to identify the maximum of cloud occurrence during the austral winter (specifically August) also detected by the CIC algorithm applied to the REFIR-PAD data. In August, a maximum in the cloud fraction is also observed in the MODIS mean L3 combined product (MCD06COSP) even if the very high value (close to 100%) looks unrealistic and biased by the observational conditions. The MODIS MCD06COSP cloud mask performance is degraded by the low insolation reaching the region around Dome C during August. This is observed also in April through September. In the same period the MODIS MYD08 and MOD08 products provides very low values of cloud fraction likely due to the low efficiency of the cloud mask algorithm based on infrared bands only.





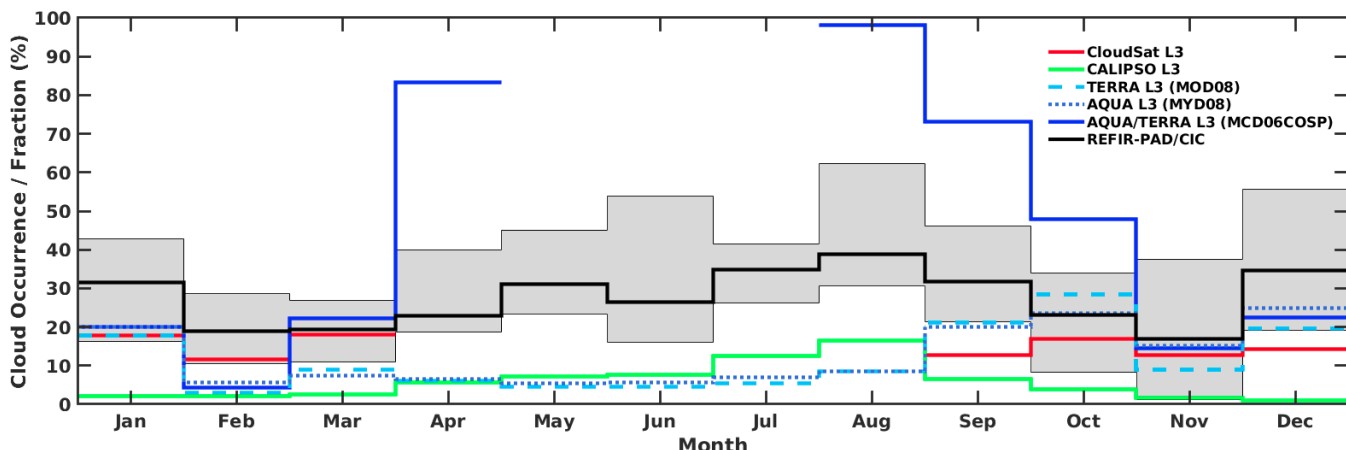

**Figure 10.** Percentage fraction of CIC monthly mean cloud occurrence (in black) compared with CloudSat L3 product (red line), CALIPSO L3 product (green line), and MODIS L3 products (solid blue line for combined AQUA and TERRA L3 product - MCD06COSP, dashed blue line for TERRA L3 product – MOD08, and dotted blue line for AQUA L3 product - MYD08). The shaded grey area indicates the minimum and maximum CIC monthly values in the interval 2012–2015.

For the months from November to March (the warm season), the CIC cloud occurrence is comparable to that found by MODIS (both MCD06COSP and MYD08/MOD08) and the CPR sensors. Nevertheless, higher percentage of cloudiness is found by the CIC algorithm with respect to the CPR. The reason is likely that the CIC is very sensitive to the identification of optically thin ice clouds which are often present in the Antarctic Plateau (Maestri et al., 2019a) and that are missed by radar measurements (Henderson et al., 2013; L'Ecuyer et al., 2008).

### 4.4 Diurnal variability of cloud occurrence

The almost continuous REFIR-PAD measurements, during the four year period provide an opportunity to investigate an hourly mean cloud occurrence. The time collocation of each CIC classification is obtained by associating each spectra to the hourly time of observation. For instance, observations performed between 1:00:00 UTC and 1:59:59 UTC are associated to the time 1:00:00 UTC. For each hour, the percentage of occurrence of each class is computed and results are reported in Figure 11. results are also presented as seasonal means.

In the austral summer (upper left panel of Figure 11), a diurnal cycle is observed and related to the hourly mean insolation, also reported in the same Figure with a black dotted curve. The clear sky occurrence is characterized by a maximum value of about 78% at around 5:00 UTC (13:00 local time). This maximum is very close in time to the maximum of insolation for the same period of the year. In the summer season, the highest percentage of occurrence of cloudiness (about 36%) is obtained during nighttime hours, that correspond to the coldest time of the day. For the other seasons, a clear diurnal cycle of the percentage occurrences is not observed. Note that for the fall and spring seasons the daily variation of the insolation is much less intense that for austral summer, and in winter is almost null. In the austral autumn (MAM, panel b) and spring (SON, panel





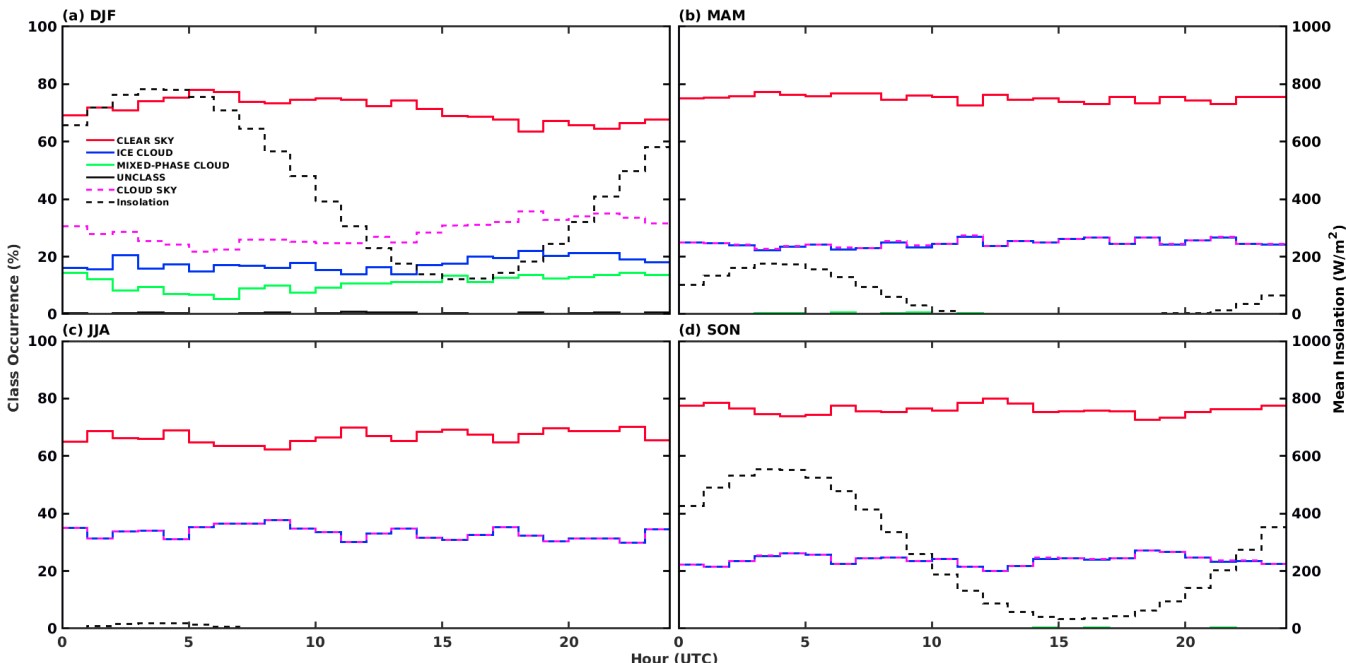

**Figure 11.** Hourly mean cloud occurrence of clear sky (red lines), ice clouds (blue lines), and mixed-phase clouds (green line). Unclassified spectra are in black and total cloud occurrence in magenta. Percentages of occurrence are provided for each season: (a) DJF, (b) MAM, (c) JJA, and (d) SON. The top of the atmosphere hourly mean insolation is also reported in $Wm^{-2}$ (black dashed line), referring to values reported on the right ordinate axis. Local time is UTC+8.

d) seasons the clouds are almost entirely composed of ice since mixed-phase clouds are very rare. In the austral winter (JJA, panel c) the insolation is close to zero and the ice cloud occurrence reaches its seasonal maximum.

In Figure 12, the hourly mean surface air temperature is plotted for the four seasons for clear sky, ice clouds, and mixed-phase clouds. The hourly mean temperatures are also presented for all-sky conditions (magenta line) and the hourly mean top of the

atmosphere insolation (dashed black curve). The all-sky hourly mean air surface temperature is driven by the diurnal cycle of insolation in summer and spring: a lag of about two hours is observed between the maximum in insolation and the maximum in temperature. The all-sky surface air temperature has a 11.2°C amplitude in summer, when the top of the atmosphere diurnal cycle of insolation is the largest. This amplitude decreases as the insolation cycle becomes weaker and it is almost null in winter.

The surface mean air temperature is higher in cloudy sky conditions (ice cloud or mixed-phase cloud) than for clear sky at all hours of the day, suggesting a positive cloud forcing at the surface level. Mean values of surface air temperature are higher in presence of mixed-phase clouds than ice clouds at all times of the day. Observations of mixed-phase clouds (green curves) are rare in autumn and spring and the data do not cover the full day in these seasons. Note that when mixed-phase clouds are present, the daily thermal amplitude is smoothed with respect to the other sky conditions. Potential explanations for this are



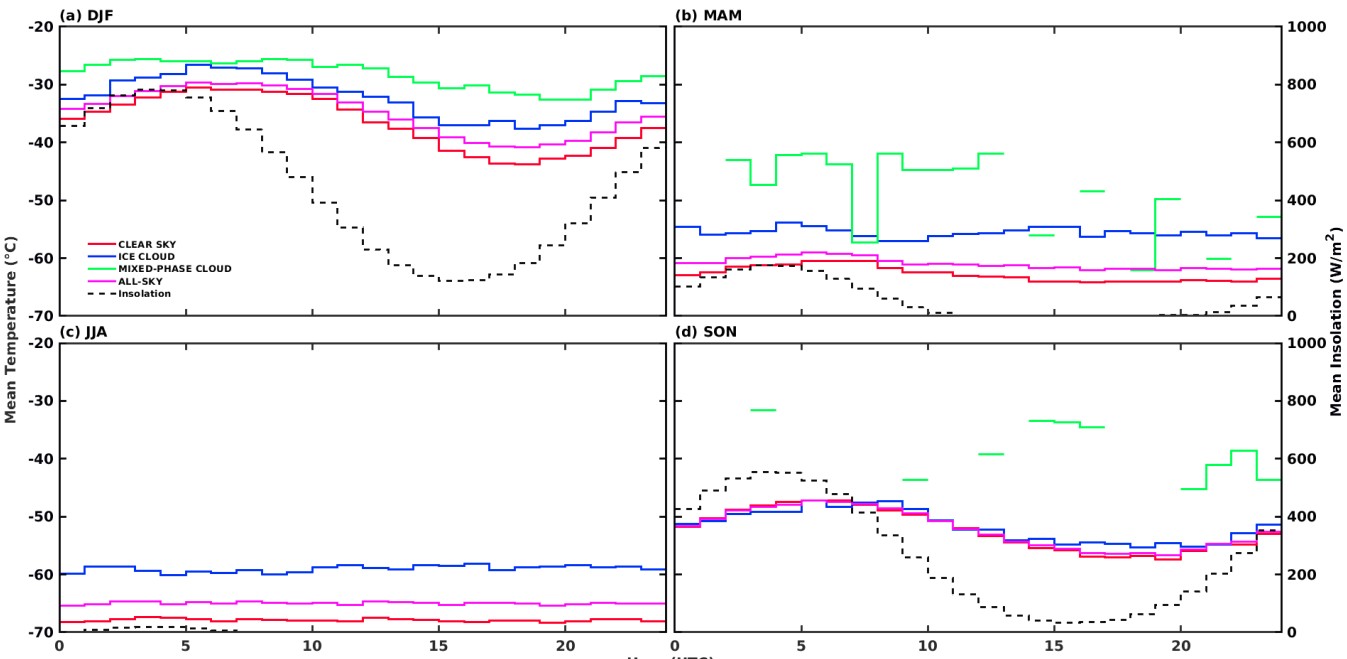

**Figure 12.** Hourly mean surface air temperature, according to the sky condition: clear sky (red line), ice cloud (blue line), mixed-phase cloud (green line), and all-sky (magenta line). The temperature is reported according to the season: (a) DJF, (b) MAM, (c) JJA, and (d) SON. The top of the atmosphere hourly mean insolation is also reported in $Wm^{-2}$ (black dashed line), and values indicated by the ordinate axis on the right. Local time is UTC+8.

this could be due to the decrease in surface insolation caused by the liquid phase clouds or perhaps caused by the larger cloud forcing of mixed phase clouds in comparison to ice clouds (Di Natale et al., 2020). The hourly mean surface temperature is larger when ice clouds are present than in clear sky conditions. This difference is, on average, larger in winter (about 9°C) and autumn (about 7°C), diminishes in summer (about 4°C) and becomes very small in spring (about 1°C). The cause of this low value needs further investigation. Possible explanations could be related to the optical thickness and position of the clouds

and/or related to the circulation of the air in the area that is not accounted for in this analysis. For the spring and summer seasons, where the insolation diurnal cycle is larger, the surface temperature difference is greater between the clear sky and ice cloud conditions for low insolation but decreases for higher insolation.

## 5   Summary and conclusion

High spectral resolution downwelling radiances at Far InfraRed (FIR) and Mid InfraRed (MIR) wavelengths are measured

by the REFIR-PAD spectroradiometer located at Dome C on the Antarctic Plateau between 2012–2015. The radiance dataset comprising spectral measurements are, for the first time, ingested by an automatic machine learning algorithm called CIC to perform single spectrum classification. CIC is developed to identify high spectral resolution observations and, in case of cloudy



scene, to perform a classification. The algorithm is computationally very fast and only requires a limited number of spectra as training set, which makes it very flexible, efficient, user-friendly, and easy to adapt to different types of sensors. For this study,

the algorithm is set-up to classify a REFIR-PAD spectrum as being a clear sky, ice cloud, or mixed-phase cloud. Mixed-phase clouds in the Antarctic Plateau are usually composed of a liquid water layer at cloud base and an ice layer close to the cloud top.

While an accurate description of clear and cloud properties is quite difficult in the Antarctic from passive measurements alone, our analysis of the REFIR-PAD data is greatly enhanced through coincident active measurents of atmospheric backscat-

tering and depolarization ratio profiles measured by a LiDAR system that is temporally co-located with the REFIR-PAD radiance measurements. The coincident LiDAR and REFIR-PAD measurements are used to obtain accurate training sets for the CIC algorithm. The training sets are formed by using a total of 202 spectra that are sufficient to characterize the large variability of the atmospheric conditions in the Antarctic Plateau region. An analysis of the LiDAR data and atmospheric vertical profiles of temperature and humidity, obtained from radiosondes launched every day at the Concordia station, is used to

separate the training sets in two macro seasons. The first is named the warm season and ranges between November and March. Three training sets, defining three different classes of spectra, are considered for the warm season: clear sky, ice cloud, and mixed-phase cloud. The second macro-season is named the cold season and corresponds to the period from April to October. For the cold season, only two classes are considered (clear sky and ice cloud) since mixed-phase clouds are rarely observed during this period due to the extremely cold atmospheric temperatures.

A number of 1726 LiDAR co-located REFIR-PAD measurements are then used to select a test set of spectra, previously classified by visual inspection using the LiDAR backscatter and depolarization ratio vertical profiles. This sample is used to test the algorithm performance, to estimate the CIC classification uncertainty, and to optimize the classification results for each class. For the optimization process, the CIC algorithm is applied to classify the test set by considering different spectral intervals. A weighted Threat Score (ThS) is used to select the optimal spectral range for the classification. Results show that

the spectral interval 380–1000 cm$^{-1}$ provides the best score due to the experimental and observational conditions. This result highlights the fundamental role of the FIR part of the spectrum to improve the process of clear/cloud identification and cloud type classification in the Antarctic.

The optimized CIC algorithm is then applied to the entire REFIR-PAD dataset from 2012 to 2015 consisting of 87758 spectra.

On average, clear sky conditions are detected in almost 72% of the cases with an associated uncertainty of the order of 1.5%. The ice cloud occurrence is about 25% and the mixed-phase clouds are identified in less than 3% of the observations. The uncertainty is 0.3% in cloudy conditions. The cloud occurrence over the Antarctic Concordia station is analysed at different temporal scales: inter-annual, seasonal, monthly, and daily variability. The inter-annual variability of total cloud occurrence spans between about 23 and 31%. A positive correlation is observed between mean air temperatures at surface level, in the

warm macro-season, and the occurrence of mixed-phase clouds. This result suggests that (a) warm temperatures are favorable for the mixed-phase clouds formation or that (b) the occurrence of warm cloud layers enhances the cloud radiative forcing at





the surface with a consequent increase in the surface temperature. Further work will help to further our understanding of mixed phase clouds in the Antarctic.

Seasonal analysis indicate that the mean total cloud occurrence varies from 23.9% in the spring (SON) to 33.2% in the cold winter season (JJA) when only ice clouds are present. In fact, most of the mixed-phase clouds are observed in summer season, where they amount to more than one third of the total clouds over Concordia station. The seasonal scene classification is analysed in accordance with meteorological parameters. Results show that highest values of surface air temperature (and relative humidity) are found in correspondence of the highest amounts of cloud for the summer, fall, and winter seasons; in the spring this relationship is minimal. The influence of the longwave radiative forcing of ice clouds on surface temperature is most observed in the winter months where the insolation is negligible. For this season, the mean surface temperature is about $-68°$C in clear sky and $-59°$C in presence of clouds. Furthermore, surface level winds from the South and South-West are more frequently observed in clear sky conditions, while in presence of ice clouds the surface wind is primarily from the South-East. When mixed-phase clouds are identified, surface winds from the East quadrant are more frequent. The mean wind intensity is about 2 ms$^{-1}$ higher in presence of ice clouds than in clear atmospheric conditions.

CIC monthly mean cloud occurrences show, on average, a maximum in August and a minimum in November. The inter-annual variability of monthly mean cloud occurrences can be very high. Noteworthy is the November case that registers a cloud occurrence variation spanning from 0 to 40% among the four years of analysis. The monthly mean data are compared with Level-3 gridded satellite products derived from the MODIS (passive imager), CALIOP (lidar) and CPR (radar) sensors. Some differences are observed among the analyzed products. In periods of intense insolation the lowest values of monthly cloud occurrence are those derived from CALIOP. Despite the low scores, CALIOP collocates the maximum cloud occurrence in winter (August) similarly to what derived by the CIC algorithm. For the months from November to March, which correspond to the warm season, the CIC cloud occurrence is larger but comparable to what is found by the MODIS and CPR sensors. The higher values detected by the CIC are probably due to the greater sensitivity of the algorithm for the identification of thin cirrus clouds and cloud layers near the surface from MIR and FIR data. The added value of both the local and continuous measurements is demonstrated. The CIC results, by exploitation of REFIR-PAD FIR and MIR spectral data available at all times during the year, provide a continuous record of cloud occurrence with excellent classification scores.

Finally, a hourly cloud occurrence analysis is performed that shows the presence of a diurnal cycle with maximum of about 36% and minimum of 22% during the austral summer that follows the hourly mean insolation. The highest cloud occurrences are observed during nighttime hours. Conversely, the season maximum of clear sky occurrence is observed in correspondence to the local noon time. For all the other seasons, diurnal cycles are not observed for either cloud or clear sky conditions. An analysis between the daily sky condition and the surface mean air temperature reveals higher surface temperatures in cloudy sky conditions, especially for mixed-phase clouds, than in clear sky for all the seasons and hours of the day. In summer, the mean surface air temperature in the presence of clouds is on average 5°C warmer than in clear sky. This difference is larger during the night but it is reduced during the day probably due to the amount of insolation. The same effect, although smaller, is observed in fall and spring due to a weaker insolation cycle. In the winter, where the insolation is almost null, the difference





between surface air temperature measured in cloudy sky and clear sky is constant at about 9°C throughout the day, which quantifies the effect of the longwave radiative forcing of the antarctic winter clouds.

The results of this work provide a basis for understanding of cloud occurrence at different time scales on the Antarctic Plateau where cloud identification and classification from satellites is challenging. The obtained results provide a useful benchmark for
satellite and model product comparisons and open the path to new investigations.

The use of FIR and MIR high spectral resolution radiances for the cloud identification and classification contributes to the preparatory studies for the Far-infrared Outgoing Radiation Understanding and Monitoring (FORUM) mission. FORUM was recently selected as the ESA's $9^{th}$ Earth Explorer mission and is scheduled for launch in 2026.

*Data availability.* The combined AQUA/TERRA L3 products are accessible at https://ladsweb.modaps.eosdis.nasa.gov/archive/allData/61/
MCD06COSP_M3_MODIS/ (Last access: $20^{th}$March 2021). The AQUA L3 products are accessible at https://ladsweb.modaps.eosdis.nasa. gov/archive/allData/61/MYD08_M3/ (Last access: $20^{th}$March 2021). The TERRA L3 products are accessible at https://ladsweb.modaps. eosdis.nasa.gov/archive/allData/61/MOD08_M3/ (Last access: $20^{th}$March 2021). The CALIPSO L3 data are accessible at https://opendap. larc.nasa.gov/opendap/CALIPSO/LID_L3_Cloud_Occurrence-Standard-V1-00/ (Last access: $25^{th}$January 2021). The CloudSat L3 data are reachable from http://www.cloudsat.cira.colostate.edu/data-products/level-aux/3f-rmcp-3s-rmcp?term=105 (Last access: $25^{th}$January
2021). The Antarctic LiDAR data are reachable at http://lidarmax.altervista.org/englidar/Antarctic%20LIDAR.php (Last access: $25^{th}$January 2021). Radiosondes and meteorological data are available at http://www.climantartide.it (Last access: $25^{th}$January 2021). REFIR-PAD radiances can be required from http://refir.fi.ino.it/refir-pad-domeC.php (Last access: $25^{th}$January 2021).

*Code and data availability.* The CIC source code version used in the present paper is available on request to the corresponding author.

*Author contributions.* TM, conceptualization. WC, DM, and TM designed the methodology, prepared and evaluated the data, performed the
cloud classification. WC, DM, MM and TM evaluated the results. WC downloaded and prepared the meteorological and satellite data. GDN, GB, LP, and MDG prepared and evaluated the REFIR-PAD and LiDAR data. All authors revised the manuscript.

*Competing interests.* The authors declare that they have no conflict of interest.

*Acknowledgements.* The present work is in preparation of the FORUM mission. FORUM related studies are supported by projects of the Italian Space Agency (ASI) and of the European Space Agency (ESA). The authors acknowledge the Antarctic Meteo-Climatological Ob-
servatory at Concordia for the meteorological data availability, from IPEV/PNRA Project 'Routine Meteorological Observation at Station Concordia'. We thank the CALIPSO Team and the Atmospheric Science Data Center at NASA Langley Research Center for archiving and



hosting CALIPSO data. We thank the CloudSat Team and the CloudSat Data Processing Center for archiving and hosting CloudSat data. We thank also the MODIS Team and the Level 1 and Atmosphere Archive and Distribution System Distributed Active Archive Center (LAADS DAAC) for making available the MODIS data.

The authors acknowledge the use of computational resources from the parallel computing cluster of the Open Physics Hub (https://site. unibo.it/openphysicshub/en) at the Physics and Astronomy Department in Bologna.





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



## List of acronyms

$\epsilon TS$ - Training Set Eigenvector Matrix

$\epsilon ETS$ - Extended Training Set Eigenvector Matrix

AMSL - Above Mean Sea Level

AWS - Automatic Weather Station

BT - Brightness Temperature

CIC - Cloud Identification and Classification

CoMPASs - Concordia Multi-Process Atmospheric Studies

CALIOP - Cloud-Aerosol LiDAR with Orthogonal Polarization

CALIPSO - Cloud-Aerosol LiDAR and Infrared Pathfinder Satellite Observation

CPR - Cloud Profiling Radar

ERB - Earth's Radiation Budget

ETS - Extended Training Set

FIR - Far InfraRed

FN - False Negative

FORUM - Far-infrared Outgoing Radiation Understanding and Monitoring

FP - False Positive

IND - INDicator function

IPEV - French Polar Institute Paul-Émile Victor (acronym from the French institute named "Institut Polaire Français Paul Émile Victor")

LiDAR - Light Detection and Ranging

MIR - Mid InfraRed

MODIS - Moderate Resolution Imaging Spectroradiometer

PCA - Principal Component Analysis

PNRA - Italian National Program for Research in Antarctica (acronym from the Italian program named "Programma Nazionale di Ricerche in Antartide")

PRANA - Radiative Properties of Water Vapor and Clouds in Antarctica (acronym from the Italian project named "Proprietà Radiative dell'Atmosfera e delle Nubi in Antartide")

REFIR-PAD - Radiation Explorer in the Far Infrared-Prototype for Applications and Development

SI - Similarity Index

SID - Similarity Index Difference

ThS - Threat Score

TN - True Negative

TP - True Positive



TS - Training Set

WMO - World Meteorological Organization