# Peer review of "Ice and Mixed-Phase Cloud Statistics on the Antarctic Plateau"

_Atmospheric Chemistry and Physics, 2021_

## Community Comment (CC2)

Overall
The presented manuscript develops a method of using REFIR-PAD spectroradiometer data to identify and track cloud properties over the Concordia station. Ancillary instrumentation is used to train a machine-learning algorithm to be applied to REFIR-PAD data.  Three years of data are then used to track cloud properties and ultimately report on cloud statistics over Concordia. In principle, the goal of presenting cloud occurrence statistics seems both reasonable and achievable given the availability of data and methods presented.  Assuming the trained data and classification scheme are correct, I see no reason to doubt the presented cloud statistic data. Furthermore, this data comes from a very data sparse part of the world and such information would be very beneficial to the community.

However, crucially, the data set used to train the algorithm must be above reproach. It is here that major concerns arise for me as a community member as I believe there are major deficiencies in the treatment of the training data. If the trained data or training method is to be doubted, the rest of the scientific value of this manuscript is degraded substantially. This is especially true given the above statement that the data come from a very sparsely sampled part of the world that would be potentially heavily relied upon to be correct and accurate.

It is my opinion that the presented manuscript contains some fairly fundamental deficiencies that need to be addressed before it should be considered by the editor for publication.

Specific Major Comments
1. While I completely recognize the paper presented does not focus on lidar, the authors seem to heavily rely on a lidar instrument, which is poorly described. It seems to me that the lidar system cannot remain as transparent as it is presented here because the reader does need to be able to evaluate the quality of the training data set. The main reference given is Palchetti et al. 2015, which is a BAMS article that seems to lack technical detail of the instrument. The Palchetti et al. 2015 paper further references a website for lidar data that seems to be defunct (at least I can't get to it on any of the computers I have tried). I am left wondering some very fundamental things about the construction of the lidar system that heavily influence its data quality. Some of the major ones (this is by no means an exhaustive list) are:
   a. Is the system coaxial or biaxial? This will affect the height range of detectable signal as well as the observed signal strength.
   b. What is the signal detection system and expected dynamic range? Does the system use photon counting or analog signals? Is the detector a photo-multiplier tube (PMT) or avalanche-photo diode (APD) or something else entirely? This affects both the height of the observable signal as well as the apparent oscillatory depolarization structure from Figure 2. For example the claim in the Palchetti et al. 2015 paper that the range is from 30-7000 m would require a

minimum signal dynamic range of 4.5 orders of magnitude (assuming a completely uniform scene). That is a tough ask even for systems that employ both analog and photon counting techniques, which introduce complexity in combining the two.

    c. What is the system field of view? This will directly affect depolarization measurements via multiple scattering.

    d. What is the laser system's divergence? Is it matched to the field of view? This combines with the above primarily relating in my mind to the possibility of observing multiple scattering.

    e. How is the system's depolarization sensitivity calibrated? Systematic effects such as internal depolarization, diattenuation, and retardance can affect all of this. For details here see for example: Biele et al. 2000, Alvarez et al. 2006, Hayman and Thayer 2009 or 2012, or Freudenthaler 2016.

    f. Do the authors use any sort of algorithm to make the backscattered signal threshold a quantitative and repeatable measure? Klett or Frenald inversions are 2 examples, which admittedly have a number of limiting assumptions required. However, it is my understanding that the authors are inspecting lidar data signal strength directly, which is neither quantitative nor repeatable. Furthermore, signal strength is complicated by alignment issues, long term degradation of optical components, atmospheric structure, and system dynamic range and design.

2. It appears that the authors are using a non-quantitative method to identify clouds using lidar data. They say on Line 213-215 that cloud identification is done by visual inspection of backscatter and depolarization profiles. If my understanding of this process is true, that is completely non-repeatable and lacks any metric whereby a reviewer or reader can either replicate or even compare results. If there is a more quantitative method to identifying clouds than what I have just described, it needs to be much more clearly stated. If this is the method, it should not really be considered quantitative at all, which undercuts the lidar data used as a standard to train the machine learning cloud identification code.

3. The authors seem to have created the following simple table to classify clouds via lidar data, which is then used to verify spectral classifications.

| Low relative signal = Clear air | High Signal
High Depol (d>0.15) = Ice |
|---|---|
| | High Signal
Low Depol (d<0.15) = Liquid or mixed phase |

Given the lack of overall description of the lidar instrument, it is not possible to evaluate if these value are reasonable. For example, the threshold between columns seems arbitrary. Furthermore, this classification scheme is very

simplified (in comparison to for example Shupe 2007 or Nott and Duck 2011) and will miss a lot of instrument related effects such as:

    a. Multiple scattering induced increase in depolarization with range
    b. Long term calibration drifts of polarization parameters
    c. Basic error propagation, e.g. is a depolarization value of $d = 0.149 \pm 1$ clear air, liquid or just bad data?
    d. Complex cloud scenes masking multilayer clouds
    e. Long term signal degradation

4. What definition of depolarization are you using? There are several in the literature, well summarized by Flynn et al. 2007 or Hayman and Thayer 2009 or 2012. Depolarization ratio vs. the Mueller matrix element d (also called depolarization) can differ by factors approaching 2. This will directly impact your ice/liquid phase classification.

5. The reference to Liou and Yang 2016 as summation of depolarization lidar is not appropriate in my opinion. There are multiple papers dating to at least Schotland et al. 1971 that are more fundamentally related to lidar such as Sassen 1991 and more recent papers such as Gimmestad 2008 or Hayman and Thayer 2012 that are complete and well known.

6. On line 283, you specify that 98% of spectra are correctly classified. That really just says that your training and test data sets are self-consistent. Furthermore, it really just says that you are pushing your reference to the lidar system. If you take the above comments numbered 2 and 3, it makes it very difficult to analyze how accurate the classification really is. Furthermore, it is impossible to replicate in any meaningful way.

7. There are a number of physical interpretations given that seem both counterintuitive and relatively easy to link to poor control of lidar data. Some examples are:

    a. Multiple scattering: You say a number of times that liquid sits below ice layers. This is counterintuitive to all the results I have seen from Arctic studies (summarized nicely by Morrison et al. 2012 and references therein). However, this is really easy to explain given the presence of multiple scattering. Even in the presence of non-depolarizing scatterers, multiple scattering can cause monotonic increases in depolarization measurements with range.

    b. You mention on line 288-289 that optically thin cloud phase is problematic to define? Without error bounds on your depolarization measurements, you cannot define how accurately you are measuring clouds, which could easily affect physical interpretations (as in the above example of $d = 0.149 \pm 1$). Second, if you are performing cloud identification (regardless of phase by visual inspection), optically thin clouds are very likely to be missed.

    c. I am really puzzled by the results in Table 5 indicating almost no observations of mixed phase clouds for 9 months out of the year. I wonder if thick liquid clouds with high occurrences of multiple scattering are being misclassified?

Specific Minor Comments
1. Line 4: Probably mean 2015
2. Line 67-68: LiDAR is first used in line 46 and should probably be defined there.
3. The capitalization of LiDAR seems odd to me. It, much like the acronym radar, is in my experience most commonly used as a word. For example, Palchetti et al. 2015 simply uses "lidar". I would suggest adopting this convention.
4. Color scheme of Figure 8. It is a minor point but using blue for ice instead of mixed phase or liquid is an odd choice to me.
5. I would also point out that Figures 2, 5, 7, 8, 10, 11 and 12 would be difficult to read for those who are red/green colorblind.

Suggested References:
1. Schotland, R.M., K. Sassen, and R.J. Stone, 1971: Observations by lidar of linear depolarization ratios by hydrometeors. J. Appl. Meteor., 10, 1011-1017.
2. Sassen, K. (1991). The Polarization Lidar Technique for Cloud Research: A Review and Current Assessment, Bulletin of the American Meteorological Society, 72(12), 1848-1866.
3. Jens Biele, Georg Beyerle, and Gerd Baumgarten, "Polarization lidar: Corrections of instrumental effects," Opt. Express 7, 427-435 (2000)
4. Alvarez, J. M., M. A. Vaughan, C. A. Hostetler, W. H. Hunt, and D. M. Winker. "Calibration Technique for Polarization-Sensitive Lidars". Journal of Atmospheric and Oceanic Technology 23.5 (2006): 683-699.
5. Connor J. Flynna, Albert Mendozaa, Yunhui Zhengb, and Savyasachee Mathurb, "Novel polarization-sensitive micropulse lidar measurement technique," Opt. Express 15, 2785-2790 (2007)
6. Shupe, M. D. (2007), A ground-based multisensor cloud phase classifier, Geophys. Res. Lett., 34, L22809, doi:10.1029/2007GL031008.
7. Gary G. Gimmestad, "Reexamination of depolarization in lidar measurements," Appl. Opt. 47, 3795-3802 (2008)
8. Matthew Hayman and Jeffrey P. Thayer, "Explicit description of polarization coupling in lidar applications," Opt. Lett. 34, 611-613 (2009)
9. Nott, G.J. and Duck, T.J. (2011), Lidar studies of the polar troposphere. Met. Apps, 18: 383-405. https://doi.org/10.1002/met.289
10. Matthew Hayman and Jeffrey P. Thayer, "General description of polarization in lidar using Stokes vectors and polar decomposition of Mueller matrices," J. Opt. Soc. Am. A 29, 400-409 (2012)
11. Morrison, H., de Boer, G., Feingold, G. et al. Resilience of persistent Arctic mixed-phase clouds. Nature Geosci 5, 11–17 (2012). https://doi.org/10.1038/ngeo1332
12. Freudenthaler, V.: About the effects of polarising optics on lidar signals and the Δ90 calibration, Atmos. Meas. Tech., 9, 4181–4255, https://doi.org/10.5194/amt-9-4181-2016, 2016.

---

## Author Comment (AC1)

Dear Reviewer and Editor,

here below our answers to your comments (reported for your convenience) are presented as a one-to-one reply and highlighted in yellow. We would like to warmly thank the Reviewer for the accurate revision of our work and in particular for raising one point concerning the discussion of the comparison of our results to L3 satellite products which gave us the possibility to better clarify the objective of the analysis.

Please note that a new version of the Article accounting for the Reviewers comments, is also attached as a reply in the discussion section.

Based on in-situ measurements, this manuscript documents ice and mixed-phase cloud statistics at the Concordia Station in the Antarctic. Various aspects of cloud statistics are discussed. A comparison with satellite L3 data product is also described.

This is a valuable study. Given the scarcity of in-situ measurements and the need to validate satellite retrievals in the polar regions, such study is also much needed by the observational community. Including far-IR, the portion of spectrum rarely used by other observational studies is another shining point of this study. However, some discussions and depictions in the text are not accurate, which I shall describe below in detail. I recommend acceptance after these issues are addressed.

Major comments:

It is well known that cloud fraction statistics from satellites hinges on multiple factors, such as the size of the field of view, the frequency used in the observation, and the detection method (active vs. passive). Besides, passive remote sensing has significant challenges in distinguishing clouds from snowy or icy surfaces over the polar region. It is good to see the authors attempted to compare REFIR-PAD cloud fraction with satellite L3 products (Figure 10) but to thoroughly and correctly interpret this figure is not trivial at all. The discussions related to Fig. 10 ignore a couple of key points: (1) cloud fraction statistics from satellite L3 products are related to footprint size, and none of the L3 products used here has the same field of view as REFIR-PAD; (2) active sensors usually can give more accurate results than passive sensors in terms of cloud occurrence, but their footprint sizes are so different that no way a real "apple-to-apple" comparison can be made.

We are aware of all the caveats highlighted by the Reviewer and for this reason we reported in the text the different sizes of gridded areas when discussing Figure 10. Similarly, the diverse measurements techniques have been mentioned as possible causes of the differences found in the comparison.

In order to make these points clearer the entire discussion of Section 4.3 is revised. The revision involves all the text of the section, and Figure 10 (here below reported for your convenience) is enriched with a right panel which visualizes the extent on the map of the L3 gridded products. Changes are applied to the abstract and the conclusions too.

[Figure]

Figure 10. Left panel: Percentage fraction of CIC monthly mean cloud occurrence (in black) compared with CloudSat L3 product (red line), CALIPSO L3 product (green line), and MODIS L3 products (solid blue line for combined AQUA and TERRA L3 product - MCD06COSP, dashed blue line for TERRA L3 product – MOD08, and dotted blue line for AQUA L3 product - MYD08). The shaded grey area indicates the minimum and maximum CIC monthly values in the interval 2012–2015. Right panel: Location of the Dome Concordia base and extension of the grid sector for Cloudsat, CALIPSO and AQUA/TERRA MODIS L3 data. Surface elevation above mean sea level is also reported.

It is also correct that the goal of the comparison has not been sufficiently described. In this regard, we added the following text in the same section (4.3):

"Monthly mean cloud occurrences/fractions derived from level 3 (L3) satellite products are also reported in the left panel of Figure 10 for the same period of time. The comparison has a twofold objective: a) to assess if the results obtained locally from the CIC/REFIR-PAD synergy can be representative of widespread region characterizing the Antarctic Plateau and b) to estimate the differences among the cloud occurrences/fractions derived from L3 satellite products around the Concordia area."

And we also added that:

"Since the L3 products of the three sensors refers to multiple extent areas of observations (of the order of tens of thousands of km2), some differences are expected not only between the ground-based measurements analysed by CIC but also among the mean values of the L3 satellite products. In particular, we note that the gridded L3 products from CALIPSO and CloudSat refer to areas characterized by important variations in surface altitude with possible consequences on cloud formation and occurrence."

And another example of discussion of the results which accounts for the Reviewer comment:

"Nevertheless, higher percentage of cloudiness is found by the CIC algorithm with respect to the CPR. The main reasons for such differences are likely due to: (1) the high CIC sensitivity to the optically thin ice clouds which are often present in the Antarctic Plateau (Maestri et al., 2019) and missed by radar measurements (Henderson et al., 2013; L'Ecuyer et al., 2008), (2) the extension of the gridded area of the CPR L3 product which comprises regions with surface elevations spanning up to 0.4 km in altitude and which might not be representative of

the Dome C conditions, and (3) the CPR coarse vertical resolution (0.5 km) which might be the cause of undetected clouds near the surface (Chan and Comiso, 2011)."

An enormous amount of effort has to be invested for data subsetting and collocation in order to make a fair comparison, which cannot be done with the L3 product directly. Figure 10 is useful and informative, but the interpretation here has to be conservative, with all caveats well described before the discussion.

We totally agree with the Reviewer on this point. In fact, this is not the goal of the analysis. We do not aim at providing any satellite products validation in the Concordia region. Such a goal would have required the exploitation of L2 products which are supplied at much higher spatial resolution and consequently can be more accurately collocated. As noted by the Reviewer this operation implies a huge effort on data selection and collocation and it goes beyond the scopes of the present paper. Anyway, we think that the plot provides insights on the accuracy of information that can be extracted from L3 datasets. The L3 products are easily accessible and provides information on the average atmospheric conditions (such as monthly mean cloud occurrence) of gridded area. When the same kind of information becomes available for a specific location independently of the satellite product, a comparison with the independent data is always instructive about the ability of gridded data to characterize a specific location.

Text is added in Section 4.3 in this regard:

"According to WMO[1], the L3 satellite products are composed of variables mapped on uniform space-time grid scales and are constructed to provide completeness and consistency for the anticipated users. These products types are frequently used to perform climate analysis and model evaluation (e.g. Stubenrauch et al., 2013; Webb et al., 2017). The assessment of their accuracy can be particularly challenging, especially in remote regions such as the Antarctic Plateau, due to the scarceness of ground-based stations that are available for products validation campaigns. For the present study, we only refer to monthly mean L3 satellite products and the comparison with CIC results is performed only in the context of the objectives described above. A validation (that is outside the scopes of the present research) should be, eventually, performed on level 2 collocated satellite products to minimize the bias due to different footprint sizes that can be otherwise very large when accounting for gridded L3 products. In practice, different data sets present specific strengths and limitations that are briefly described below."

For example, the abstract stated, "A comparison of monthly mean 15 results with cloud occurrences/fractions derived from level 3 satellite products, from passive and active sensors, emphasizes the difficulties of satellite observations in the Antarctic region and highlights the ability of the CIC/REFIR- PAD synergy to identify multiple cloud conditions and studying their variability at different time scales.", which I think is not a fair statement given all the reasons mentioned above.

True. The abstract is modified as follows: "Monthly mean results are compared to cloud occurrence/fraction derived from gridded (Level-3) satellite products, from both passive and active sensors. The differences observed among the considered products and the CIC results are analysed in terms of footprint sizes and sensors' sensitivities to cloud optical and

geometrical features. The comparison highlights the ability of the CIC/REFIR-PAD synergy to identify multiple cloud conditions and study their variability at different time scales."

Other comments:

Figure 2c, red spectra, I am surprised to see the clear-sky spectrum here has a peak at $CO_2$ band center (~667 cm-1) as low as 150 K BT. Since this is a surface measurement looking up, the BT at the $CO_2$ band center should be close to the temperature in the lower troposphere or even in the boundary layer. Thus, it cannot be so low. The cloudy spectra here, as well as the clear-sky spectra in Figure 5, look all reasonable to me. Thus, this 150K BT peak at the $CO_2$ band center in Figure 2c needs to be examined and explained.

The brightness temperature peak at 15-um is a calibration artefact due to the noise amplification (due to calibration process) present at this wavenumber caused by the strong air absorption inside the interferometric path. Since the noise is not shown in the figure, for a better clarity, we will remove this spectral point in the revised figures (both figure 2 and 4). The noise plots of REFIR-PAD measured in the Antarctic campaign are reported in https://doi.org/10.5194/amt-12-619-2019.

Please note that radiances at wavenumber from 620 to 670 cm-1 (in the $CO_2$ band, with the inclusion of the Q-branch at band center 667 cm-1) are not ingested by the CIC algorithm and thus the indicated bad calibration channel does not affect the classification results. This is also the case of others bad calibrations points at FIR channels below 300 cm-1 and above 1200 cm-1, caused, respectively, by strong absorption from $H_2O$ rotational lines, and by the absorption of the Mylar beam-splitter. These bands too are not ingested by CIC which is applied in the 380-620 and 670-1000 cm$^{-1}$ spectral intervals.

The information about the exclusion of the 620-670 cm-1 spectral interval was provided in previous papers such as the one describing the CIC algorithm [Maestri et al., 2019] and the one accounting for its first application to airborne data [Magurno et al., 2020]. Since the information is missing in the present paper, a new sentence is added for completeness:

"Note that, as discussed in Maestri et al. (2019b) and Magurno et al. (2020), the spectral interval 620-670 cm−1 is excluded by the analysis."

Line 130: The impact of multiple scattering within liquid clouds on the depolarization ratio can be included as justification for the 15% depolarization threshold.

Yes. We added the following comments: "In this study a depolarization ratio of 0.15 is used as a threshold for the discrimination of the liquid water clouds and ice clouds over the Concordia Station. The value accounts for possible increases due to multiple scattering effects as discussed below."

and

"The 15% depolarization ratio value is selected to account for the impact of multiplescattering within liquid clouds. It is observed that in presence of mixed-phase clouds the depolarization ratio shows very small values at cloud base, characteristics of liquid spheres, and increases towards values typical of ice crystals near the cloud top. An increase

is, in part, intrinsically related with liquid water layers, where multiple scattering determines a depolarization that gradually increases with the depth of penetration, in the lidar backscatter. For this reason, in some conditions, the phase of the upper part of the cloud cannot be unambiguously defined based on the analysis of the depolarization ratio profile only. Nevertheless, the presence of liquid phase at bottom is unequivocally identified and the cloud is categorized as mixed-phase."

Line 154: These classes are not explicitly mentioned in the abstract, please list them there.

Now the abstract reports:
"The CIC algorithm is optimized for Antarctic sky conditions and results in a total hit rate of almost 0.98, where 1.0 is a perfect score, for the identification of the clear sky, ice and mixed-phase clouds classes."

Table 3: Misclassifications are in the hit rate column. Please make it clearer that these are misclassifications through labeling or reformatting the table.
A new column is added to Table 3 accounting for the misclassified data

Figure 7: The caption mentions unclassified spectra, but there does not seem to be any in the figure.

True. An automatic software for plotting was used which accounts for all the possibility. A new figure is generated with a correct caption.

Line 345: The authors mention that a subset of the long-term data is used for training and testing. Is this different from the training data and testing data mentioned in the preceding text? If so, please state. If not, then please indicate why the algorithm is retrained and reoptimized.

Rephrased:
"A total of 87960 REFIR-PAD spectra are analysed from the dataset spanning over the time range 2012-2015. From this set, only 202 spectra are used for training the CIC algorithm, and the other 87758 are ingested by the CIC to evaluate the cloud occurrence over the Concordia station."

---

## Author Comment (AC2)

Dear Reviewer and Editor,

here below our answers to your comments (reported for your convenience) are presented as a one-to-one reply and highlighted in yellow. We would like to warmly thank the Reviewer for the detailed revision of our work. In particular, we noted that the Reviewer's comments are in some cases scientific considerations (and not only criticisms) which open for new discussion and ideas. We really appreciated this approach.

Please note that a new version of the Article accounting for the Reviewers comments, is also attached as a reply in the discussion section.

This is a very interesting study of cloud type and frequencies of occurrence observed from MIR and FIR spectra over the Antarctic Plateau. The study is fairly well written and informative; I recommend acceptance after these comments have been considered by the authors.

Relatively minor questions:

1. It would be quite interesting to provide an analysis of the surface-based lidar measurements for cloud occurrence, thermodynamic phase, cloud heights and layers, even if the measurements are limited to a height of about 7km. The lidar analysis would be complementary to the REFIR-PAD analysis, if there was enough data collected.

We agree with the Reviewer about the scientific interest of the proposal. Nevertheless, we believe that the proposed work should, in case, be the topic of a new research and not included in the present paper. At the moment, an automatic classification algorithm based on surface lidar measurements is not available. Thus, we visually inspected all the available lidar backscatter and depolarization profiles to classify first 202 REFIR-PAD spectra used as elements of the training sets and an additional set of 1726 REFIR-PAD spectra used as test set for CIC. The application of the CIC algorithm to the test set results in a 98% of correctly classified spectra meaning that there is an excellent agreement between the lidar and radiometric information in this regard. For this reason, it is expected that a classification based on surface-lidar data only (that is in any case unavailable at the moment) would provide very similar results (to what obtained by CIC) if applied to the entire 4 years dataset. Note also that the lidar range in altitude is not a limitation in this case since the Concordia base is placed at 3,3 km a.s.l. and the Tropopause height is usually lower than 10 km in that region.

2. I have two questions about the lidar data:

a. does multiple scattering above a liquid layer or at the cloud top impact the interpretation of the results? There is some evidence for this in the upper panel of Figure 7.

As observed by the Reviewer, the liquid layers show an increase in the depolarization ratio as we move from cloud bottom to cloud top. This is observed both in Figure 2 and Figure 7. This increase is intrinsically related with liquid water layers, where multiple scattering determines a depolarization that increases with the depth of penetration, in the lidar backscatter. Nevertheless, the liquid water layer is neatly identified by the lidar due to the very small depolarization ratio observed at cloud base. For a classification point of view, the identification of liquid particles in the layer is the key information which makes the observed cloud to pertain to a specific category (mixed-phase in our nomenclature) that is different from 'pure' ice clouds. The two categories (ice and mixed phase clouds) show peculiar radiometric features in the REFIR-PAD spectrum which are captured by the CIC classificator. For this reason, we believe that multiple scattering above the liquid layer does not affect the classification results.,

In response to the Reviewer comment the sentence at line 138 has been deleted and substituted by: "The 15% depolarization ratio value is selected to account for the impact of multiple scattering within liquid clouds. It is observed that in presence of mixed-phase clouds the depolarization ratio shows very small values at cloud base, characteristics of liquid spheres, and increases towards values typical of ice crystals near the cloud top. An increase is, in part, intrinsically related with liquid water layers, where multiple scattering determines a depolarization that gradually increases with the depth of penetration, in the lidar backscatter. For this reason, in some conditions, the phase of the upper part of the cloud cannot be unambiguously defined based on the analysis of the depolarization ratio profile only. Nevertheless, the presence of liquid phase at bottom is unequivocally identified and the cloud is categorized as mixed-phase."

b. Is there evidence of ice particles falling through the base of the liquid layer? How often does this happen? This question arose when I read lines 290-296, and studied Figure 7 and related text. This is a very interesting point. The frequency of occurrence of this process has not been quantified yet. M. Del Guasta (one of the authors), who is responsible for lidar data, reports that falling ice from mixed-phase clouds is a common situation at Dome C. It is also noted that, in the summer season, liquid water cloud layers with no associated precipitation are observed close to the surface (in the firsts hundreds of meters). A sentence describing this possible condition is reported at the end of the section indicated by the Reviewer. Note that, from a classification point of view, the presence of a liquid layer is sufficient to imply the conditions for 'mixed-phase' clouds.

"Another common situation is the presence of falling ice from mixed-phased cloud layers, as shown in the mid panel of Figure 2 between 18 and 20 UTC. Typically, the quantity of the precipitating ice crystals is very small and the CIC algorithm is able to capture the radiometric signal from the upper liquid water layer as it will be shown in the case reported in Figure 7."

3. In the paragraph beginning on line 437, I am puzzled by the lack of cloud fraction information in the winter (dark) months for the combined Terra and Aqua MODIS cloud product. If the information is available for the MODIS product from each of the Terra and Aqua platforms, there must be a problem with the combined data product. This seems to be something that the MODIS cloud team has to resolve. Suggest leaving out the combined Terra and Aqua data product until it has been resolved.
A new Figure 10 is generated accounting for two different MODIS cloud products. This is explained in the text as follows:
"Two types of MODIS L3 products are used in this study: MCD06COSP and MYD08/MOD08. The first one combines the observations from both AQUA and TERRA platforms (MCD06COSP_L3, MODIS Atmosphere Science Team, 2020). This product is based on a cloud mask which uses bands at visible and infrared wavelengths. Thus, the cloud fraction information (solid blue in Figure 10) is not available in the Antarctic region during the dark season (between May and July). The second product is derived from each MODIS sensors on platforms separately (MYD08 for AQUA, and MOD08 for TERRA, MODIS Atmosphere Science Team, 2017). The MOD08/MYD08 L3 product is based on a cloud mask which exploits infrared bands when in absence of solar illumination. In this case, the monthly mean cloud fraction is available for all the seasons (dashed and dotted blue curves in Figure 10, for MODIS TERRA and AQUA L3 products, respectively"

The combined data product (MCD06COSP) was developed by the MODIS team in the context of the project "Level 3 Atmosphere for CFMIP (Cloud Feedback Model Intercomparison Project) Observation Simulator Package or COSP". The COSP (Bodas-Salcedo et al., 2011) products have been extensively used to evaluate and validate climate trends (Saponaro et al., 2020; Zhang et al., 2019), and for this reason we think that it should be considered in the comparison.

The product MCD06COSP has some differences with respect to individual L3 satellite product (MOD08/MYD08). As said, in our case the main difference is that the Cloud Fraction of the combined product (MCD06COSP) is computed by using the Cloud Mask flags from L2 products of each satellite for daytime only while in case of MOD08/MYD08 L3 products the cloud mask exploits infrared bands when in absence of solar illumination.

Note also that the entire Section 4.3 is improved and the differences among the diverse satellite L3 products and CIC results are analyzed in terms of different area extensions and sensor sensitivities to cloud features.

References:
Bodas-Salcedo, A., et al. (2011). COSP: Satellite simulation software for model assessment. Bulletin of the American Meteorological Society, 92(8), 1023-1043. doi: 10.1175/2011BAMS2856.1.
Saponaro, Giulia, et al. (2020). Evaluation of Aerosol and Cloud Properties in Three Climate Models Using MODIS Observations and Its Corresponding COSP Simulator, as Well as Their Application in Aerosol-Cloud Interactions. Atmospheric Chemistry and Physics, vol. 20, no 3, p. 1607-1626. doi:10.5194/acp-20-1607-2020.
Zhang, Yuying, et al. (2019). Evaluation of Clouds in Version 1 of the E3SM Atmosphere Model With Satellite Simulators. Journal of Advances in Modeling Earth Systems, vol. 11, no 5, p. 1253-1268. doi:10.1029/2018MS001562.

Other comments:

Title: "on Antarctic Plateau" —> "on the Antarctic Plateau"
Done

Line 114: does 1928 refer to the number of REFIR-PAD spectra that are collocated with LiDAR measurements?
Yes. Rephrased: "A set of 1928 REFIR-PAD spectra are co-located with LiDAR measurements."

Line 126: I think there's a problem with the reference "Sassen and yu Hsueh". Should be Sassen and Hsueh.
The reference is updated.

Line 142: Radiosondes Vaisala RS92 —> Radiosondes (Vaisala RS92)
Done

Line 176: are arranged —> are prepared
Done

Line 188: generally small cloud —> generally low cloud
Done

Line 199: that can be different —> which can be different
Done

Line 228: "window wavenumbers, that results in a very"—> window wavenumbers, and the measurements can have very
Done

Line 286: "if for the 8.3" —> if for 8.3
Done

Line 302: (c) —> or (c)
Done

Line 345: as TS —> for training
Following also suggestions from reviewer #1, this sentence was rephrased to: "From this set, only 202 spectra are used for training the CIC algorithm, and the other 87758 are ingested by the CIC to evaluate the cloud occurrence over the Concordia station."

Lines 361-363: A third possibility is suggested by the authors but not included in the sentence: The temperature and mixed-phase cloud correlation could indicate that warm temperatures are favorable for mixed-phase clouds formation or that the presence of warm liquid clouds implies a stronger cloud forcing at the surface and, consequently, an increase in the temperature values near the ground. The third possibility is warm air advection of moisture. If the authors agree on this point, this third possibility should be included here and also in the Conclusion section.
The possibility is mentioned: "Another favorable condition for liquid cloud formation consists in the advection of air from warmer and more humid regions such as the Ross Sea and Southern Ocean."

Line 391: in correspondence of cloud sky conditions —> when clouds are present
Done

Line 412: in presence of different —> for different
Done

Lines 409-410: please mention where the winds from the NE originate to provide some potential insight as to the origination of the moist layer that is being advected over the Plateau.
The sentence is rephrased: "Note (see back to Figure 1) that South and West directions at the Concordia station point to the inner Antarctic Plateau, where the drier air is supposedly found. Otherwise, the South-East and East directions are towards the Ross Sea and the Southern Ocean which are characterized by warmer and more humid air. The correlations are far from being conclusive since the upper level winds and the back trajectories of the air masses have not been analyzed yet."

Line 434: by both satellites platforms —> by both satellite platforms
Done

Line 439-440: Thus —> For some reason... Note: the MODIS cloud mask should always have a result regardless of solar illumination because it includes infrared measurements.
See the answer to point 3 of the "Relatively Minor Comments", which deals with the same argument

Line 456: in case of —> in the case of
Done

Line 460: In months in which —> In the months where

==Done==

Line 460: CALIOP products in green —> CALIOP products as shown in green
==Done==

Line 461: maximum of —> maximum in
==Done==

Line 497: in presence of —> in the presence of
==Done==

Lines 499-500: Potential explanations for this are this could be due... —> Reasons for this could include...Actually, this entire sentence is a bit awkward and should be reworked.
==The sentence is re-phrased: "Note that when mixed-phase clouds are present, the daily thermal amplitude is smoothed with respect to the other sky conditions. The main reason for this could be related with the averagely larger optical thickness of liquid water clouds with respect to ice clouds \citep{dinatale20} which implies a decrease in surface insolation and thus a dumping of the diurnal cycle of surface temperature due to the reduced solar warming."==

Line 512: classification —> classifications
==Done==

Line 515: set up —> optimized
==The sentence is modified as follows: "For this study, the algorithm is arranged and optimized to classify a REFIR-PAD spectrum as clear sky, ice cloud, or mixed-phase cloud."==

Line 525: sets in two —> sets into two
==Done==

Lines 545-547: could include warm air advection as a third option
==The sentence is now: "This result suggests that (a) warm temperatures due to meteorological conditions (including warm and humid air advection) are favorable for the mixed-phase clouds formation or that (b) the occurrence of warm cloud layers enhances the cloud radiative forcing at the surface with a consequent increase in the surface temperature. Further work is needed for a better identification of the key atmospheric conditions and understanding of the physical processes driving to mixed-phase clouds formation in the Antarctic."==

Line 564: intense insolation —> higher insolation
==Done==

Line 565: CALIOP collocates the —> CALIOP data indicates that the
==Done==

Line 566: similarly to what derived —> similar to what is derived
==Done==

Line 572: a hourly —> an hourly
==Done==

Line 572: with maximum —> with a maximum
==Done==

Line 573: and minimum —> and a minimum
Done

Line 579: but it is reduced —> but smaller
Done

---

## Author Comment (AC3)

Dear Reviewer and Editor,

here below our answers to your comments (reported for your convenience) are presented as a one-to-one reply and highlighted in yellow. We would like to warmly thank the Reviewer for the revision of our work and in particular the detailed analysis of the spectral radiance signal.

Please note that a new version of the Article accounting for the Reviewers comments, is also attached as a reply in the discussion section.

This paper presents a comprehensive analysis of four years of far-infrared and mid- infrared spectral measurements of downwelling radiation at the Dome C site in Antarctica. Nearly 88,000 spectra comprise the database. These spectra are analyzed with a machine learning code to identify scenes as clear or comprised of ice-phase or mixed-phase clouds. The method of training the machine learning algorithm using coincident lidar data is described.

The infrared spectral observations are automated and are taken throughout the year. Analyses of the occurrence of the different scene types are presented in various ways (by year, by time of year (season)). Comparisons against observations of cloud type made by satellites are also presented. The paper also presents an analysis of the occurrence of cloud type against the meteorological conditions.

The paper is very exhaustive in its analyses and I would recommend publication after these minor comments are addressed.

Line 101 – are the effects of the air in the 1.5 m chimney significant at any wavelength? My team found it necessary to account for the "chimney effect" in analysis of our ground- based, uplooking data. See https://doi.org/10.1016/j.jqsrt.2015.10.017 and http://dx.doi.org/10.1016/j.jqsrt.2017.04.028

We are aware of the "chimney effect" on localized spectral intervals characterized by large gaseous optical depths and short photon paths: such as the $CO_2$ Q-branch and the water vapor most absorbing lines associated to rotational transitions in the FIR, typically below 400 cm-1 for the considered atmospheric conditions. This effect is related to the calibration process as explained in the replies to the next 2 comments regarding figure 2 and 5. In previous research, such as in Rizzi et al 2016 (See Section 4 and 5 in this regard.), we have applied corrections to chimney effect since the goal was the study of the radiance signal along the REFIR-PAD spectrum. Also, the effect is considered when we retrieve the atmospheric profiles and cloud properties as in Di Natale et al. (2020).

In the present research, we don't use the spectral interval from 620-670 cm-1 (a note is added to the text) and the CIC optimization selects the 380-1000 cm$^{-1}$ band (again with the exclusion of the 620-670 cm-1) as the reference interval for the classification. Thus, the chimney effect does not interfere with the classification analysis.

Di Natale, G.; Bianchini, G.; Del Guasta, M.; Ridolfi, M.; Maestri, T.; Cossich, W.; Magurno, D.; Palchetti, L. Characterization of the Far Infrared Properties and Radiative Forcing of Antarctic Ice and Water Clouds Exploiting the Spectrometer-LiDAR Synergy. *Remote Sens.* 2020, *12 (21)*, 3574. https://doi.org/10.3390/rs12213574

Rizzi, R., C. Arosio, T. Maestri, L. Palchetti, G. Bianchini, and M. Del Guasta (2016), One year of downwelling spectral radiance measurements from 100 to 1400 cm$^{-1}$ at Dome Concordia: Results in clear conditions, J. Geophys. Res. Atmos., 121, 10,937–10,953, http://dx.doi.org/10.1002/2016JD025341

Line 116 – what is meant by (1928)?

Rephrased: "A subset of REFIR-PAD data, comprising 1928 spectra, is co-located with LiDAR measurements."

Figure 2 – The 150 K brightness temperature in the red (clear sky) curve is interesting. This is at the very core of the 15-um band, the strongest part of the band, so the line should be saturated almost immediately within the atmosphere, and thus, 150 K appears to be too small. Is there a suspected reason for this anomalous feature? Or is it perhaps a microwindow saturating in the polar summer mesosphere where the temperatures can be well below 150 K? These are January (summer) spectra, so the polar mesosphere is quite cold at that time.

Actually, the brightness temperature peak at 15-um is a calibration artefact due to the noise amplification (due to calibration process) present at this wavenumber caused by the strong air absorption inside the interferometric path. Since the noise is not shown in the figure, for a better clarity, we will remove this spectral point in the revised figure. The noise figures of REFIR-PAD measured in the Antarctic campaign are reported in https://doi.org/10.5194/amt-12-619-2019. As said, the band 620-670 cm$^{-1}$ is excluded by the CIC identification process which relies only on brightness temperatures spanning over the 380-620 and 670-1000 cm$^{-1}$ bands. This is clarified in the new version of the paper.

Figure 5 – Similarly, the 300 K brightness temperature seem a bit high for such a saturated part of the spectrum. In the summer season the polar stratopause temperature can approach 300 K, but in the winter season this seems too warm. In addition, the brightness temperatures near 1300 cm-1 and the standard deviations approaching 350 K are non-physical. To what extent does the cloud classification system (CIC) depend on these regions of the spectrum in which the brightness temps appear to be incorrect?

As for the 150 K peak at 15-um (667 cm$^{-1}$), these features are due to the noise amplification caused by the low efficiency of the interferometer due to strong absorption present inside the interferometer at specific wavenumbers. Below 300 cm$^{-1}$ they are caused by $H_2O$, at 667 cm$^{-1}$ by $CO^2$ and around 1100 and 1270 cm$^{-1}$ by the absorption of the Mylar beam-splitter. Note that also these spectral intervals are not accounted for in the CIC analysis.

Figures 8 and 11. The legends in these figures are difficult to read as the font is very small. I am looking at the figures on my computer screen that projects each manuscript page at full size. The labeling of the axes of these figures is also difficult to read. Please re- plot these figures with larger axis labels and please use a larger font on the figure legends.

The Figures are generated again using larger font sizes for the axes labels and legends.

---

## Author Comment (AC4)

**Overall**: This paper presents results from a unique and valuable dataset. The two main contributions are cloud classification of this dataset into clear skies, ice cloud, and mixed phase cloud, and an algorithm that can quickly do this classification (which is presumably applicable elsewhere). However, we believe there are a number of serious problems with this paper. Most importantly, there is insufficient evidence that the authors are classifying cloud phase. Instead, it appears likely that they are grouping views into 3 types: 1) clear sky, 2) colder, optically thinner clouds, and 3) warmer, optically thicker clouds. Given this, it is not clear what value the algorithm adds to the literature, given that other methods exist that can classify phase that also classify optical depth and hydrometeor effective radius. The authors need to determine and report what they are actually classifying views into, e.g. using simulated data. More details follow.

Respectfully, we disagree with the reader's deductions. Regarding the two points ("problems") raised:

- The category Mixed phase (in the training set, in the test set and in the entire dataset) comprises clouds with optical depths spanning from about 0.1 to the order of some unities.
- The article deals with the application of the identification and classification method. The algorithm's features and edge on other methods are discussed in previous articles and only summarized here.

More details follow.

**A better review of the literature and comparison to existing methods are needed**

Referencing of the lidar instrument and how phase is determined is insufficient.

More details are provided in a new version of the article, also in response to another community comment.

Referencing of recent work on Antarctic cloud properties and similar cloud property retrievals is insufficient. Reading this paper, it would seem that there have been no surface-based studies of Antarctic clouds after 2012. The authors should reference recent papers by Lachlan-Cope et al 2016, Silber et al 2018, Lubin et al 2020, etc.

The current version of the article contains references to studies presented in 2019 and 2020. Anyway, the suggested literature will be taken into consideration for the final version of the work (we already knew some of the suggested works) and we are grateful for the advice. New references will be eventually inserted in the article in accordance with the goal and focus of the present work.

Machine learning concepts need to be referenced. Due to a complete lack of such references, it is unclear what are established methods (PCA, confusion matrix, hit rate, etc) and what was invented by the authors. E.g. are there references for the method of using a test set and an extended test set? Summing subtracted eigenvectors? Such references would be very helpful to fill in gaps and help understand what is novel.

**Suggestion accepted.**

The paper should compare this new method to existing methods for retrieving cloud phase from infrared radiances. For example, they reference Cox et al 2014, who retrieve cloud properties from Arctic infrared radiances, but do not compare to this work. They should also reference and discuss comparison to Rowe et al 2019 & Lubin et al 2020, which includes development and application of a cloud property retrieval, including phase, to clouds over McMurdo, Antarctica. Simulated datasets exist which could be used for an inter-comparison of methods. See, e.g. Cox et al 2016, Earth System Science Data, 8(1), 199–211.

We don't perform a retrieval but a cloud identification and classification. The cited references are all regarding retrievals. The CIC classification methodology was tested against synthetic and real data in previous publications and master thesis. In the paper by Di Natale et al. (2020) the reader can evaluate the results of a retrieval process considering of a large subset of the same dataset analyzed here.

**Examination of the data in a real-world context is needed**

The authors report the common occurrence of cloud with a liquid base and an ice layer at the top, which is contrary to what has been reported previously, both in the Arctic and Antarctic. This difference from previous work calls for some justification. This also underscores the need for a better explanation of the lidar design and methodology for determining cloud phase. What is meant by determining cloud layers from lidar by "human intervention?" Is this objective and repeatable? Why can't it be automated? Overall, using lidar as truth is not properly justified.

The procedure describing the usage of lidar data for the definition of the 3 classes will be better explained in the final version of the paper. See also reply to Reviewers 1 and 2. Moreover, the definition of mixed phase cloud will be better detailed. The lidar data are perfectly appropriate for the definition of the 3 classes of observation (i.e Ricaud 2020, 2017), especially for the Antarctic Plateau region conditions in which optically and geometrically thin clouds are very frequent.

The authors use Principal Component Analysis (PCA), but they never explore, plot, or discuss the associated eigenvalues and eigenvectors. The retrieval is blind in the sense that it does not take into account the atmospheric state in terms of temperature,

humidity, CO2 concentration etc. This would be ok if it was shown that the retrieval works without taking these into consideration, including some exploration of how it works, but this has not been done.

There is a paper (see references) describing the CIC algorithm and its main features. One of the strengths of the algorithm (as described in the reference) is that it is based on the signal only and doesn't need any other ancillary information to perform the classification. The present work is an application of the method and not a repetition of the description of the algorithm.

It should be noted that almost all the variance, and thus the strongest PCs, will be associated with cloud temperature and optical depth, not phase. Which PCs are associated with phase? Why use all PCs believed to be above the noise level?

We agree with the generic statement of the reader. Nevertheless, a one to one relation among the PCs and cloud features cannot be generalized. The reader is invited to review the algorithm methodology and in particular the metric defining the classification. We never analyze a spectrum singularly, but only in addition to all the training set elements.

It seems likely that the classification is not based on cloud phase at all, but rather that scene views are subdivided into: 1) clear sky, 2) colder, optically thinner clouds, and 3) warmer, optically thicker clouds. They call category 2 "ice" and category 3 "mixed phase." It is possible these classifications are often correct, since ice clouds tend to be optically thinner and colder, and liquid clouds tend to be optically thicker and warmer on the Antarctic Plateau. However, this needs to be characterized, addressed and discussed, including errors and caveats. Several lines of evidence support the idea that they are not classifying cloud phase but rather optically thick and warm vs optically thin and cold clouds. First, looking at Fig. 2, it is unlikely that it is possible to determine phase from the green spectrum. This spectrum looks saturated, which means phase will have no influence on it - that is, there is no information about phase. It does, however, indicate that the cloud is optically thick. The authors could assess for which cases phase cannot be retrieved, using simulated spectra. Instead, are all such cases classified as "mixed phase" by the algorithm?

The deduction is erroneous and somehow unjustified.

What shown in Figure 2 are just two examples of spectra. We agree that maybe two different spectra should be selected as examples to avoid possible confusion. In the new version of the paper this point will be made clearer.

The training sets of ice and mixed-phase clouds are both composed of thin and thick clouds. Below one example of a thin ice cloud and one example of a thin mixed-phase cloud. Note that the spectra reported in the figure below are from the same day considered in Figure 2.

See also Di Natale et al. 2020 which shows cloud properties retrievals (including OD) from a subset of the CIC classified spectra presented in this work. In the cited paper the range of values over which the cloud optical depths span can be evaluated both for ice and mixed phase clouds

Second, as the authors point out, it has been shown that the far IR is critical for determining phase. Yet Fig. 6 suggests that a wavenumber range that excludes the far IR altogether would be equally good as one that includes it: the threat score is close to 1 for a range of just above 560 cm-1 to ~1020 cm-1. Indeed, the authors find the best range to be 540-1020 cm-1 for mixed phased clouds (it is unclear how they determine this), excluding essentially all of the far IR.

The methodology for the selection of the best range is described in the text. Probably the plots do not evidence enough the advantage of exploiting the FIR that is, anyway, clear from the numerical result concerning the Threat Score of the test set (for all the classes). A 3-d plot could be used in the new version of the article which better highlights the advantage of using FIR channels down to 380 cm-1. The scale can also be adapted to highlight the enhancement in the threat score. Note that an increase of few cents in the threat score means a large increase in the number of spectra correctly classified when dealing with the entire dataset (87960)

Typically, mixed-phase clouds are associated to more humid conditions than ice clouds and also, as described in the new text, to precipitation of thin ice crystals. For these reasons, the inclusion of the smallest wavenumbers (associated to the less transparent part of the FIR) did not bring significant enhancement in the classification.

Note that the classification of the dataset spectra is performed using the interval producing the best ThS over all classes: 380-1000 cm-1.

Third, in the cold macro-season the algorithm does not retrieve cloud phase at all; instead all clouds are assumed to be ice.

**Correct**

Given the above, the authors should report the results of testing their method on simulated data, as has been done for other methods in the literature. This would allow them to test whether they truly have a cloud phase categorizer or if they are categorizing by cloud temperature / optical thickness. They could also determine and define characteristics of each category in terms of temperature, optical depth and phase ranges. This would also allow exploration of how errors propagate.

Tests against simulated (and measured) data were performed in the papers introducing the CIC (see literature). We also recall to the reader that the CIC is the official cloud classificator in the ESA FORUM E2E simulator and it is severely tested everyday by many scientists in the community.

Also, it is mentioned in the text that sensitivity tests. applied to synthetic stratified clouds with constant total OD, provide different classifications according to the relative amount of liquid/ice water content. The tests prove that CIC does not rely on a single parameter (i.e. optical depth) but on the entire spectrum characterization.

In the current paper we don't perform any retrieval. Nevertheless, the results of the classification include thin clouds both in the ice and mixed phase category. A large subset of the identified spectra is analyzed in the recent work by Di Natale et al. (2020). Di Natale et al. (2020) perform a cloud optical and microphysical retrieval. Please check the results. For example, you can evaluate the retrieved OD: it ranges from about 0.1 to about 4 in case of ice (figure 9) and about 0.1 to 10 in case of mixed phase (figure 10).

The authors ignore previous work on the temperature dependence of the singlescattering parameters (SSPs) of liquid water, which indicate that the SSPs of supercooled liquid water are intermediate between those of liquid and ice (Rowe et al 2013 and 2020 and references therein). In particular, Rowe et al (2020) indicates that uncertainties are large in the far IR.

We don't ignore it. The paper was brought to our attention by the reader via email 2 days before we submitted the present article. Anyway, we recall to the reader that the classification is a "discretized" result. We are aware that there are conditions "in the middle" not only due to the SSP, but also because of the physical structures of the observed scenes. It is noted, in the new version of the article, that what we call mixed phase cloud frequently occurs as a precipitating ice layer plus a thicker layer with low depolarization ratio values plus an upper layer with increasing depolarization ratio.

**Questions and Concerns about Methodology and use of Machine Learning**

The authors need to justify why they used the method they developed. It is not clear why PCA is used, or why the SID is used. Why isn't a simpler method tried, or at least compared to, to justify the more complicated method used?

The methodology is accurately described in previous papers. We regret that the details of the methodology applied are not clear to the reader and in particular the CIC ability in identifying optically thin clouds that are missed by simplistic (i.e. based on thresholds) methodologies. We also note that in Maestri et al. (2019) a methodology (CCREF) based on a combined linear discriminant analysis and support vector machine method is used on a similar dataset. Please refer to the cited literature. Some effort will be performed to better resume the CIC potentialities in the new version of the current paper.

Fig. 5 suggests that only one wavenumber is needed to distinguish cloudy from clear skies. Such a cloud mask has been reported in the literature but is not referenced or noted here (e.g. Weaver et al 2017, Atmos. Meas. Tech., 10, 2851–2880, 2017, Appendix). Classification using a single wavenumber would be sufficient for all of the cold macro-season data. Why is a considerably more complicated method used?

Fig 5 is used for illustrative goals only and shows the Training set mean BTs and their standard deviations. The figure just demonstrates what the reader is stating: "ice clouds tend to be optically thinner and colder, and liquid clouds tend to be optically thicker and warmer on the Antarctic Plateau". Nevertheless, the mean spectra are not used in any part of the classification process. Otherwise, the plotted quantities suggest that a large variability exists within the elements composing the classes (note the magnitude of 1 standard deviation).

As far as very simple methodologies based on single channels or BT difference thresholds, it has been demonstrated that they fail in detecting optically thin clouds which are very frequent in the considered experimental conditions.

To distinguish ice cloud from mixed-phase cloud, how many PCs are needed? Is PCA justified? Also, it seems odd to first divide cases into clear sky vs ice cloud and confusing that these each include mixed-phase. Why not divide first to clear and cloudy? Then subdivide cloudy into ice and mixed-phase. Such important details are left unexplored by the authors.

The number of the PCA used comes from the empirical function, called the factor indicator function (IND), defined by Malinokowski (1977, 2002) and reported in Turner et al (2006). It is usually of the order of ten in our case and it represents the number of PCs that allows the correct identification of the elements of each training set. About the flow of the comparison, a team in one country may decide to do things quite differently than a team in another country. There is much to be learned from comparisons and discussion. The authors use PCA to remove noise (Eqns 3-4) using an established method. However, Antonelli et al (2004, J Geophys Res 109, D23102), who should be referenced, state that the size of the training set should be greater than the number of spectral elements (M>N) to most accurately reconstruct the atmospheric signal and most efficiently remove noise. Here it appears that M<<N. How does this affect the noise reduction and signal reconstruction?

We know the work of Paolo, nevertheless we don't want to accurately reconstruct the full radiometric signal of each selected spectrum. Otherwise, CIC defines a metric based on the changes in the main PCs characterizing a set of spectra (the training set spectra) when a new element (the analyzed spectrum) is added to the set. The evaluation of the change (the modified information content) is assessed always as a comparison with respect to the change obtained when a different training set is considered. This is a change of metric with respect to previous methods. Please refer to references cited in the text.

Antonelli et al (2004) also state that if some spectra are not well-represented by the set of spectra used for noise reduction, a larger number of PCs may be needed to properly represent those spectra. This seems likely to be the case when the input spectrum is not a member of the training set in Eq. (6). How is this handled and how does it impact the results?

This is one point that justify the methodology used. Please, see the two answers above.

The authors reduced the dimensionality of the observations by modifying the spectral interval of the test set members and re-running the algorithm. Given that the authors are already using PCA, and that PCA is typically used for dimensionality reduction, why isn't PCA used for this dimensionality reduction?

PCA, as well as other techniques, could be used to reduce the dimensionality in this sense. We have used linear discriminant analysis in Maestri et al. (2019) to this goal. However, such a method would select specific wavenumbers along the spectrum, according with the highest eigenvectors elements. This is not what we are interested in at the moment. For general purpose and to maintain the methodology simplicity, we want to select continuous portions of the spectrum. This led the procedure to remove the smallest and largest measured wavenumbers which are affected by the largest instrument noise. In such a way the maximum amount of information is anyway passed to the PCA used in the CIC.

Furthermore, it is not good practice to use the test set to select features (wavenumbers) to use. Using the test set to optimize the algorithm exaggerates the accuracy of the method and can lead to overfitting. Model development should be done using training or validation sets. See, e.g. Ripley, B.D. (1996) Pattern Recognition and Neural Networks,

Cambridge: Cambridge University Press, p. 354. The data with known labels should be split into training, validation, and testing sets. The testing set should be held apart and only used to estimate the accuracy of the method. None of the training, testing, or validation data should then be included in the analysis. The authors need to clarify which data is being used in each step and ensure they are following established practice.

We are aware of the procedure described by the reader and we have used it in the past. Nevertheless, the selection process of the input wavenumber interval cannot be properly defined as a hyper-parameter setting. It simply reduces the amount of channels ingested by the CIC. This is not a parameter that directly affects the behaviour of the algorithm. The algorithm operates exactly the same way independently of the selected interval. In this sense, a proper validation set is, therefore, not necessary.

Indeed, splitting what is currently defined as "test set" into two distinct groups of equal size (a "validation set" and a "test set") leads to the same exact results reported in the current version of the paper. We can provide these results if needed. The same wavenumber interval is selected on the validation set (380-1000 cm-1), and the same hit rates are found for the test set, implying the same results on the entire dataset.

**More detail is needed to allow the analysis to be repeated.**

It is not clear how the authors handle erroneous data points. One of the reviewers pointed out that a data point at the center of the CO2 band is erroneous, at 667 cm-1. This is typical with such instruments because calibration is impossible at such wavenumbers (see Rowe et al 2011, Optics Express, 19(6), 5451–5463, and Optics Express 19(7), 5930-5941). There are many other erroneous brightness temperatures evident - for example, none of the BTs below 200 cm-1 appear useable, as well as many between 200 and ~350 cm-1, where BTs are very high. How did the authors handle such points in their analysis? Were they included or omitted? The authors should briefly explain the instrument error characterization and point to a reference with more detail.

Correct. The bad calibration points fall in spectral regions that are not considered in the analysis. This is explained in the answers to reviewers 1 and 3 and implemented in the new text.

The algorithm description could use some clarification. The development should proceed linearly from training to testing to implementation. It seems that what is meant by the input spectrum on line 164 varies; this needs to be clarified. For example, it seems the SIDs and the CSIDs are developed from the training set first (to get Fig. 4)? How is this done?

The request of clarification is not clear to us. The development proceeds exactly as described. The CSID definition is part of the optimization process. See the text and references.

Finally, the utility of this algorithm seems likely to be specific to the unique conditions on the Antarctic Plateau. The authors should discuss whether it would be applicable elsewhere.

We respectfully note that the CIC methodology, in less than 3 years, has been applied to:

- Spectral radiance simulations over the globe
- The airborne TAFTS data
- The airborne ARIES data
- Satellite FORUM simulated radiances within the ESA End2End simulator (two different instances of sensor are considered plus another ideal sensor)
- the ground based REFIR-PAD

It is all in the cited literature. At the moment we are applying the CIC to IASI and FIRMOS data.

**References**

Di Natale, G.; Bianchini, G.; Del Guasta, M.; Ridolfi, M.; Maestri, T.; Cossich, W.; Magurno, D.; Palchetti, L. Characterization of the Far Infrared Properties and Radiative Forcing of Antarctic Ice and Water Clouds Exploiting the Spectrometer-LiDAR Synergy. Remote Sensing, 2020, 12 (21), 3574. https://doi.org/10.3390/rs12213574.

Maestri, T.; Arosio, C.; Rizzi, R.; Palchetti, L.; Bianchini, G.; Del Guasta, M. Antarctic Ice Cloud Identification and Properties Using Downwelling Spectral Radiance From 100 to 1,400 Cm –1. J. Geophys. Res. Atmos., 2019, 124 (8), 4761–4781. https://doi.org/10.1029/2018jd029205.

Maestri, Tiziano; Cossich, William; Sbrolli, Iacopo, Cloud identification and classification from high spectral resolution data in the far infrared and mid-infrared, «ATMOSPHERIC MEASUREMENT TECHNIQUES», 2019, 12, pp. 3521 - 3540. DOI: 10.5194/amt-12-3521-2019

Magurno, D.; Cossich, W.; Maestri, T.; Bantges, R.; Brindley, H.; Fox, S.; Harlow, C.; Murray, J.; Pickering, J.; Warwick, L.; Oetjen, H. Cirrus Cloud Identification from Airborne Far-Infrared and Mid-Infrared Spectra. Remote Sens. 2020, 12(13), 2097; https://doi.org/10.3390/rs12132097.

Malinowski, E. R. Determination of the number of factors and the experimental error in a data matrix. Anal. Chem., 1977, 49 , 612–617. <a href="https://doi.org/10.1021/ac50012a027">https://doi.org/10.1021/ac50012a027</a>

Malinowski, E. R. Factor Analysis in Chemistry. 3d ed. Wiley and Sons, 2002, 414 pp.

Ricaud, P.; Bazile, E.; del Guasta, M.; Lanconelli, C.; Grigioni, P.; Mahjoub, A. Genesis of Diamond Dust, Ice Fog and Thick Cloud Episodes Observed and Modelled above Dome C, Antarctica. Atmos. Chem. Phys., 2017, 17 (8), 5221–5237. https://doi.org/10.5194/acp-17-5221-2017.

Ricaud, P.; Del Guasta, M.; Bazile, E.; Azouz, N.; Lupi, A.; Durand, P.; Attié, J. L.; Veron, D.; Guidard, V.; Grigioni, P. Supercooled Liquid Water Cloud Observed, Analysed, and

Modelled at the Top of the Planetary Boundary Layer above Dome C, Antarctica. Atmos. Chem. Phys., 2020, 20 (7), 4167 4191. https://doi.org/10.5194/acp-20-4167-2020.

Turner, D. D.; Knuteson, R. O.; Revercomb, H. E. Noise Reduction of Atmospheric Emitted Radiance Interferometer (AERI) Observations Using Principal Component Analysis, J. Atmos. Ocean. Tech., 2006, 23, 1223-1238. https://doi.org/10.1175/JTECH1906.1

---

## Author Comment (AC5)

Overall

The presented manuscript develops a method of using REFIR-PAD spectroradiometer data to identify and track cloud properties over the Concordia station. Ancillary instrumentation is used to train a machine-learning algorithm to be applied to REFIR-PAD data. Three years of data are then used to track cloud properties and ultimately report on cloud statistics over Concordia. In principle, the goal of presenting cloud occurrence statistics seems both reasonable and achievable given the availability of data and methods presented. Assuming the trained data and classification scheme are correct, I see no reason to doubt the presented cloud statistic data. Furthermore, this data comes from a very data sparse part of the world and such information would be very beneficial to the community.

Thanks for the interest in our work. We have used four years of data (2012-2015).

However, crucially, the data set used to train the algorithm must be above reproach. It is here that major concerns arise for me as a community member as I believe there are major deficiencies in the treatment of the training data. If the trained data or training method is to be doubted, the rest of the scientific value of this manuscript is degraded substantially. This is especially true given the above statement that the data come from a very sparsely sampled part of the world that would be potentially heavily relied upon to be correct and accurate. It is my opinion that the presented manuscript contains some fairly fundamental deficiencies that need to be addressed before it should be considered by the editor for publication.

We expect that the community members ask for clarifications and provision of details when their goal is a better understanding of the obtained results. The present research is the result of a huge effort by a community of scientists who work in this field with professionality from several years. Thus, we are fully available to respond to requests going in this direction and not affected by any preconceptual judgment.

Specific Major Comment

1. While I completely recognize the paper presented does not focus on lidar, the authors seem to heavily rely on a lidar instrument, which is poorly described. It seems to me that the lidar system cannot remain as transparent as it is presented here because the reader does need to be able to evaluate the quality of the training data set. The main reference given is Palchetti et al. 2015, which is a BAMS article that seems to lack technical detail of the instrument. The Palchetti et al. 2015 paper further references a website for lidar data that seems to be defunct (at least I can't get to it on any of the computers I have tried). I am left wondering some very fundamental things about the construction of the lidar system that heavily influence its data quality.

The lidar system will be better described in the next version of the paper and new references will be added.
However, the description of the same lidar system was presented in many other articles (Di Natale et al., 2020; Maestri et al., 2019; Rizzi et al. 2016; Tomasi et al., 2015), and also in 3 previous works published in ACP (Ricaud et al., 2020, 2017; Chen et al., 2017). In all these publications the description of the lidar was considered satisfactory by the community. We are surprised that the reader has completely missed all the relative recent articles even the ones that are more focused on the lidar data.
Concerning the website indicated in Palchetti et al. (2015), the reader is right since the provided link was incorrectly typed.

The website is always active at the following link:
http://lidarmax.altervista.org/englidar/Antarctic%20LIDAR.php
(accessed on 17 May 2021).

Some of the major ones (this is by no means an exhaustive list) are:

Here below there is a long list of technical details. We agree with the reader that some of these details could be included in the next version of the article. Nevertheless, we would like to keep the focus on the identification and classification process of spectral radiances and the discussion of the results rather than the lidar.

a. Is the system coaxial or biaxial? This will affect the height range of detectable signal as well as the observed signal strength.

The system is biaxial, with 10 cm off-axis.
Please note that the Dome C site is located at almost 3300 m a.s.l and clouds are usually found between surface and 2 km above surface level.

b. What is the signal detection system and expected dynamic range? Does the system use photon counting or analog signals? Is the detector a photo-multiplier tube (PMT) or avalanche-photo diode (APD) or something else entirely? This affects both the height of the observable signal as well as the apparent oscillatory depolarization structure from Figure 2. For example the claim in the Palchetti et al. 2015 paper that the range is from 30-7000 m would require a minimum signal dynamic range of 4.5 orders of magnitude (assuming a completely uniform scene). That is a tough ask even for systems that employ both analog and photon counting techniques, which introduce complexity in combining the two.

- LIDAR Detection: Hamamatsu PMT, analog mode.  Automatic avoidance of signal saturation from g.l.  upward through laser power modulation
- Signal averaging over 1000 laser shots
- True: signal dynamic range of 4.5 orders of magnitude is required assuming a completely uniform scene. But this is not the case: the molecular atmosphere at 7000m is not detected, just cirrus clouds are (with a scattering ratio of at least 10).  7000 m is the upper limit for detection: Concordia cirrus are well below 4000 m altitude. Moreover, in case of low clouds laser power is automatically reduced, so that a signal compression is always obtained. The figure of 4.5 magnitudes thus doesn't apply to this instrument.

c. What is the system field of view? This will directly affect depolarization measurements via multiple scattering.

FOV is approx. 2 mrad full angle.

d. What is the laser system's divergence? Is it matched to the field of view? This combines with the above primarily relating in my mind to the possibility of observing multiple scattering.

The nominal laser aperture is 1 mrad full angle.

e. How is the system's depolarization sensitivity calibrated? Systematic effects such as internal depolarization, diattenuation, and retardance can affect all of this. For details here see for example: Biele et al. 2000, Alvarez et al. 2006, Hayman and Thayer 2009 or 2012, or Freudenthaler 2016.

The calibration is obtained by inserting a lambda/2 plate at the laser exit (in order to have e roughly 50% power on both polarizations). In absence of clouds, a measurement of the ratio between the two (p,s) output signals (averaged over a 1000 m window) is obtained. The two pmts are exchanged of place (keeping everything else unchanged in the acquisition chain) and a second ratio (same window) is obtained. The geometric mean of the two ratios provides a measure of calibration ratio, insensitive with respect to changes in the atmosphere and laser power.

f. Do the authors use any sort of algorithm to make the backscattered signal threshold a quantitative and repeatable measure? Klett or Frenald inversions are 2 examples, which admittedly have a number of limiting assumptions required. However, it is my understanding that the authors are inspecting lidar data signal strength directly, which is neither quantitative nor repeatable. Furthermore, signal strength is complicated by alignment issues, long term degradation of optical components, atmospheric structure, and system dynamic range and design.

No algorithm was used for this type of study: Concordia LIDAR is used as a range-finder for cloud base, top, vertical extension, time evolution and water phase (liquid/solid) from depolarization. Background/offset subtraction only is applied.  No quantitative LIDAR data about backscatter, extinction, and else are in fact given in the paper. The ratio of offset-corrected signals is extremely reliable in providing our simple information. Methods like the one by Klett are unapplicable as automatic procedures, as the reader correctly suggests, and quite unreliable in complex atmospheres.

Clouds are very complicated objects. The parameters that define a cloud span over a large range of values. The lidar quicklooks are available to the reader (please visit the website). The list of the times of lidar measurements used in the study to define the training sets can be provided as additional material.  For all the considered cases the atmospheric structure associated with the presence or the absence of a cloud are neatly identifiable. The selection of the training set elements was performed accurately choosing the observations and avoiding the most complicated cases.

2. It appears that the authors are using a non-quantitative method to identify clouds using lidar data. They say on Line 213-215 that cloud identification is done by visual inspection of backscatter and depolarization profiles. If my understanding of this process is true, that is completely non repeatable and lacks any metric whereby a reviewer or reader can either replicate or even compare results. If there is a more quantitative method to identifying clouds than what I have just described, it needs to be much more clearly stated. If this is the method, it should not really be considered quantitative at all, which undercuts the lidar data used as a standard to train the machine learning cloud identification code.

The reader is partially repeating the same question posed at point 1.f.  See reply above.

Please note that we identified clearly three different atmospheric conditions through the lidar measurements: the clear sky, the ice cloud and a category called mixed phase.

The mixed phase category is better described in the next version of the paper.
Mixed phase clouds are characterized by a layer with small values of the depolarization ratio at cloud base (less than 15%), characteristics of liquid spheres. The depolarization ratio increases towards values typical of ice crystals near the cloud top.  An increase is, in part, intrinsically related with liquid water layers, where multiple scattering determines a depolarization that gradually increases with the depth of penetration, in the lidar backscatter. For this reason, in some conditions,

the phase of the upper part of the cloud cannot be unambiguously defined based on the analysis of the depolarization ratio profile only. Nevertheless, the presence of liquid phase at bottom is unequivocally identified and the cloud is categorized as mixed-phase. Moreover, a common situation is the presence of falling ice from mixed-phased cloud layers, as shown in the mid panel of Figure 2 between 18 and 20 UTC. Typically, the quantity of the precipitating ice crystals is very small and the CIC algorithm is able to capture the radiometric signal from the upper liquid water layer as it will be shown in the case reported in Figure 7. For a classification point of view, the identification of liquid particles in the layer is the key information which makes the observed cloud to pertain to a specific category (mixed-phase in our nomenclature) that is different from 'pure' ice clouds.

The two categories (ice and mixed phase clouds) show peculiar radiometric features in the REFIR-PAD spectra which are captured by the CIC classificator. For this reason, we believe that multiple scattering above the liquid layer does not affect the classification results.

3. The authors seem to have created the following simple table to classify clouds via lidar data, which is then used to verify spectral classifications.

| Low relative signal = Clear air | High Signal
High Depol (d>0.15) = Ice |
|---|---|
| | High Signal
Low Depol (d<0.15) = Liquid or mixed phase |

Given the lack of overall description of the lidar instrument, it is not possible to evaluate if these value are reasonable. For example, the threshold between columns seems arbitrary. Furthermore, this classification scheme is very simplified (in comparison to for example Shupe 2007 or Nott and Duck 2011) and will miss a lot of instrument related effects such as:
a. Multiple scattering induced increase in depolarization with range
b. Long term calibration drifts of polarization parameters
c. Basic error propagation, e.g. is a depolarization value of $d = 0.149 \pm 1$ clear air, liquid or just bad data?
d. Complex cloud scenes masking multilayer clouds
e. Long term signal degradation

Many authors use the threshold found by Intrieri et al. (2002) as a reference which uses a depolarization ratio of 0.11 to discriminate between ice and liquid water particles. We used the value (0.15) indicated by Sassen (1991). The lidar system is now better described and the reader can evaluate if the threshold is reasonable.

Note that the schemes proposed by Shupe (2007) exploits co-located measurements from lidar, radar, microwave radiometer, and temperature vertical profiles to classify clouds, which are not available in our case. In the experimental conditions encountered at Dome C clouds are very thin and composed of small particles that makes the proposed methodology totally ineffective.

Moreover, note that the use of the depolarization ratio to define the cloud phase has been used in many papers, such as the cited Nott and Duck, (2011).

4. What definition of depolarization are you using? There are several in the literature, well summarized by Flynn et al. 2007 or Hayman and Thayer 2009 or 2012. Depolarization ratio vs. the Mueller matrix element d (also called depolarization) can differ by factors approaching 2. This will directly impact your ice/liquid phase classification.

The depolarization ratio used is the simple signal depolarization (cross polarized total signal/ parallel polarized total signal *100), after background subtraction.

As the scattering ratio R of most Concordia cirrus exceeds 10, this means that a typical signal depolarization of 30% reported in the paper could in fact correspond to a cloud depolarization (cross polarized cirrus signal/ parallel polarized cirrus signal *100) of 33%,

Furthermore, the possible ambiguity between liquid phase clouds and oriented ice plates is avoided at Dome C by operating the lidar 4° off-zenith (Ricaud et al., 2020).

5, The reference to Liou and Yang 2016 as summation of depolarization lidar is not appropriate in my opinion. There are multiple papers dating to at least Schotland et al. 1971 that are more fundamentally related to lidar such as Sassen 1991 and more recent papers such as Gimmestad 2008 or Hayman and Thayer 2012 that are complete and well known.

Major comment?
We provided a general definition. We think that the reference book is appropriate, but we are available to add a new reference.

6. On line 283, you specify that 98% of spectra are correctly classified. That really just says that your training and test data sets are self-consistent. Furthermore, it really just says that you are pushing your reference to the lidar system. If you take the above comments numbered 2 and 3, it makes it very difficult to analyze how accurate the classification really is. Furthermore, it is impossible to replicate in any meaningful way.

Respectfully, we disagree with the reader's comment that, in principle, can be applied to every classification procedure based on automatic learning.
Moreover, the reader is assuming that the cases included in the test set are perfectly mirroring what is included in the training set.
The lidar data is used to define 3 distinct classes as clarified above. These classes correspond to 3 typical lidar backscatter and depolarization ratio observations in Antarctica. Moreover, they also correspond to specific radiometric signals in the FIR and MIR that are neatly recognized by the CIC algorithm in most of the cases (but not all).
See also replies to other community reader.

Differently to other methodologies, the classification is easily repeatable. We can provide the list of the times of the lidar and refir-pad measurements contained in the training set as additional material to the article. Note that CIC simply ingests the training set spectra and the dataset spectra and performs the classification without any other tuning. We are not aware of others methodology which work so straightforward. In the past we have used neural network and support vector machine methodologies.

7. There are a number of physical interpretations given that seem both counterintuitive and relatively easy to link to poor control of lidar data. Some examples are:
a. Multiple scattering: You say a number of times that liquid sits below ice layers. This is counterintuitive to all the results I have seen from Arctic studies (summarized nicely by Morrison et al. 2012 and references therein). However, this is really easy to explain given the presence of multiple scattering. Even in the presence of non-depolarizing scatterers, multiple scattering can cause monotonic increases in depolarization measurements with range.

We have improved the description of the mixed-phase class and included a discussion of the multiple scattering effect that goes in the direction of the reader's comment. See also replies to reviewer 1 and 2.

b. You mention on line 288-289 that optically thin cloud phase is problematic to define? Without error bounds on your depolarization measurements, you cannot define how accurately you are measuring clouds, which could easily affect physical interpretations (as in the above example of d = 0.149 ± 1). Second, if you are performing cloud identification (regardless of phase by visual inspection), optically thin clouds are very likely to be missed.

Yes, the sentence should be re-phrased. The IR radiance signal in presence of cloud approaches the clear sky radiance signal as the cloud optical depth becomes thinner.
The sensitivity of CIC to thin cirrus clouds has been tested in previous studies such as Maestri et al. (2019) and Magurno et al. (2020).

c. I am really puzzled by the results in Table 5 indicating almost no observations of mixed phase clouds for 9 months out of the year. I wonder if thick liquid clouds with high occurrences of multiple scattering are being misclassified?

We think that we made clear in the text that mixed-phase clouds are not considered in the cold season (i.e. from April to October included).
The low occurrence of mixed-phase clouds is in accordance with results from Listowski et al. (2019), being near to zero in MAM, JJA, and SON.

Specific Minor Comments
1. Line 4: Probably mean 2015
Corrected

2. Line 67-68: LiDAR is first used in line 46 and should probably be defined there.
No. In that case the word LIDAR is used in the expression of the acronym for CALIOP.

3. The capitalization of LiDAR seems odd to me. It, much like the acronym radar, is in my experience most commonly used as a word. For example, Palchetti et al. 2015 simply uses "lidar". I would suggest adopting this convention.
We adopt simply "lidar", as suggested.

4. Color scheme of Figure 8. It is a minor point but using blue for ice instead of mixed phase or liquid is an odd choice to me.
The colors (red for clear sky, blue for ice clouds, and green for mixed-phase clouds) are kept the same in all figures to facilitate the analysis for the readers.

5. I would also point out that Figures 2, 5, 7, 8, 10, 11 and 12 would be difficult to read for those who are red/green colorblind.

Yes, the reader is right, but we don't have a solution for this at the moment.

Suggested References:

1. Schotland, R.M., K. Sassen, and R.J. Stone, 1971: Observations by lidar of linear depolarization ratios by hydrometeors. J. Appl. Meteor., 10, 1011-1017.
2. Sassen, K. (1991). The Polarization Lidar Technique for Cloud Research: A Review and Current Assessment, Bulletin of the American Meteorological Society, 72(12), 1848-1866.
3. Jens Biele, Georg Beyerle, and Gerd Baumgarten, "Polarization lidar: Corrections of instrumental effects," Opt. Express 7, 427-435 (2000)
4. Alvarez, J. M., M. A. Vaughan, C. A. Hostetler, W. H. Hunt, and D. M. Winker. " Calibration Technique for Polarization-Sensitive Lidars". Journal of Atmospheric and Oceanic Technology 23.5 (2006): 683-699.
5. Connor J. Flynna, Albert Mendozaa, Yunhui Zhengb, and Savyasachee Mathurb, "Novel polarization-sensitive micropulse lidar measurement technique," Opt. Express 15, 2785-2790 (2007)
6. Shupe, M. D. (2007), A ground-based multisensor cloud phase classifier, Geophys. Res. Lett., 34, L22809, doi:10.1029/2007GL031008.
7. Gary G. Gimmestad, "Reexamination of depolarization in lidar measurements," Appl. Opt. 47, 3795-3802 (2008)
8. Matthew Hayman and Jeffrey P. Thayer, "Explicit description of polarization coupling in lidar applications," Opt. Lett. 34, 611-613 (2009)
9. Nott, G.J. and Duck, T.J. (2011), Lidar studies of the polar troposphere. Met. Apps, 18: 383-405. https://doi.org/10.1002/met.289
10. Matthew Hayman and Jeffrey P. Thayer, "General description of polarization in lidar using Stokes vectors and polar decomposition of Mueller matrices," J. Opt. Soc. Am. A 29, 400-409 (2012)
11. Morrison, H., de Boer, G., Feingold, G. et al. Resilience of persistent Arctic mixed- phase clouds. Nature Geosci 5, 11–17 (2012). https://doi.org/10.1038/ngeo1332
12. Freudenthaler, V.: About the effects of polarising optics on lidar signals and the Δ90 calibration, Atmos. Meas. Tech., 9, 4181–4255, https://doi.org/10.5194/amt-9-4181-2016, 2016.

References

Chen, X.; Virkkula, A.; Kerminen, V.-M.; Manninen, H. E.; Busetto, M.; Lanconelli, C.; Lupi, A.; Vitale, V.; Del Guasta, M.; Grigioni, P.; et al. Features in Air Ions Measured by an Air Ion Spectrometer (AIS) at Dome C. Atmos. Chem. Phys., 2017, 17 (22), 13783–13800. https://doi.org/10.5194/acp-17-13783-2017.

Di Natale, G.; Bianchini, G.; Del Guasta, M.; Ridolfi, M.; Maestri, T.; Cossich, W.; Magurno, D.; Palchetti, L. Characterization of the Far Infrared Properties and Radiative Forcing of Antarctic Ice and Water Clouds Exploiting the Spectrometer-LiDAR Synergy. Remote Sensing, 2020, 12 (21), 3574. https://doi.org/10.3390/rs12213574.

Intrieri, J. M. An Annual Cycle of Arctic Cloud Characteristics Observed by Radar and Lidar at SHEBA. J. Geophys. Res., 2002, 107 (C10). https://doi.org/10.1029/2000jc000423.

Listowski, C.; Delanoë, J.; Kirchgaessner, A.; Lachlan-Cope, T.; King, J. Antarctic Clouds, Supercooled Liquid Water and Mixed Phase, Investigated with DARDAR: Geographical and Seasonal Variations. Atmos. Chem. Phys., 2019, 19 (10), 6771–6808. https://doi.org/10.5194/acp-19-6771-2019.

Maestri, T.; Arosio, C.; Rizzi, R.; Palchetti, L.; Bianchini, G.; Del Guasta, M. Antarctic Ice Cloud Identification and Properties Using Downwelling Spectral Radiance From 100 to 1,400 Cm −1. J. Geophys. Res. Atmos., 2019, 124 (8), 4761–4781. https://doi.org/10.1029/2018jd029205.

Maestri, T.; Cossich, W.; Sbrolli, I. Cloud Identification and Classification from High Spectral Resolution Data in the Far Infrared and Mid-Infrared. Atmos. Meas. Tech., 2019, 12 (7), 3521–3540. https://doi.org/10.5194/amt-12-3521-2019.

Magurno, D.; Cossich, W.; Maestri, T.; Bantges, R.; Brindley, H.; Fox, S.; Harlow, C.; Murray, J.; Pickering, J.; Warwick, L.; et al. Cirrus Cloud Identification from Airborne Far-Infrared and Mid-Infrared Spectra. Remote Sensing, 2020, 12 (13), 2097. https://doi.org/10.3390/rs12132097.

Nott, G. J.; Duck, T. J. Lidar Studies of the Polar Troposphere. Met. Apps, 2011, 18 (3), 383–405. https://doi.org/10.1002/met.289.

Ricaud, P.; Bazile, E.; del Guasta, M.; Lanconelli, C.; Grigioni, P.; Mahjoub, A. Genesis of Diamond Dust, Ice Fog and Thick Cloud Episodes Observed and Modelled above Dome C, Antarctica. Atmos. Chem. Phys., 2017, 17 (8), 5221–5237. https://doi.org/10.5194/acp-17-5221-2017.

Ricaud, P.; Del Guasta, M.; Bazile, E.; Azouz, N.; Lupi, A.; Durand, P.; Attié, J.-L.; Veron, D.; Guidard, V.; Grigioni, P. Supercooled Liquid Water Cloud Observed, Analysed, and Modelled at the Top of the Planetary Boundary Layer above Dome C, Antarctica. Atmos. Chem. Phys., 2020, 20 (7), 4167–4191. https://doi.org/10.5194/acp-20-4167-2020.

Sassen, K. The Polarization Lidar Technique for Cloud Research: A Review and Current Assessment. Bull. Amer. Meteor. Soc., 1991, 72 (12), 1848–1866. https://doi.org/10.1175/1520-0477(1991)072<1848:tpltfc>2.0.co;2.

---

## Author Response (AR1)

Dear Editor and Reviewers,

as you noticed we have already replied to all the Referees and Community Readers comments.

As indicated in the one-to-one replies to the Referees the text has been largely modified to account for all the requests.

Also, the Readers comments have been accounted for and new details have been added to the text. With respect to what already described in the one-to-one replies, we have applied the following modifications:

- The introduction is improved, and new references are considered. This was a strong suggestion by the Community.
- The lidar system is better described and technical details are provided as requested by the Community readers.
- A subsection (2.1) is introduced. The process used to identify mixed-phase clouds through lidar data is presented and detailed. References are added.
- Figure 2 is re-plotted. In particular, the mixed-phase cloud spectrum used as example is changed in order to avoid confusion. In this way it is clearer that mixed-phase clouds do not coincide with optically thick layers.
- Bad calibration points in Figure 2 and 5, falling in the range 380-1000 $cm^{-1}$, are deleted.
- The color scale of Figure 6 is modified to better highlight the maxima in each panel.
- Figure 10 is re-plotted without accounting for the comparison of the MODIS MCD06COSP products since data in winter months were missing. The text is modified accordingly. MOD08 and MYD08 are used instead.

Best regards, Tiziano Maestri and all the co-authors

---

## Author Response (AR2)

Reply to Reviewers

Dear Editor and Reviewers,

==Here below (in yellow) a one-to-one reply to all the suggestions and comments provided by the Reviewers. Again, we warmly would like to thank all of you for the detailed work provided which greatly helped improving the quality of our research.==
==Best regards, Tiziano Maestri and co-Authors.==

Reviewer #1 (B. Baum)

Here are some minor grammatical suggestions for the authors to consider:
==We noticed that the Reviewer referred to line numbering in the latex-diff document. All suggestions are accepted.==

Line 165: Suggest rewording the sentence beginning with "It is not infrequent the occurrence of..." to something like this: The occurrence of precipitating ice crystals from mixed-phase cloud layers is not infrequent.

Lines 332-333: as it will be shown —> as will be shown

Line 473: scarceness —> scarcity

Line 496: from MODIS sensor is showed Figure 10 —> from the MODIS sensors are shown in

Line 503: in right panel —> in the right panel

Lines 524-525: products provides —> products provide

Line 528: higher percentage —> a higher percentage

Line 532: which comprises —> that encompasses

Line 564: averagely larger — relatively larger

Line 565: dumping —> dampening

Line 637: benefits of the shortwave —> benefits from the shortwave

==All suggestions are accepted and modifications applied.==

Reviewer 2 (Xianglei Huang)

Suggestions for revision or reasons for rejection (will be published if the paper is accepted for final publication)

The authors have done a good job addressing my comments. The revised manuscript is much improved in terms of presentation. I commend the authors for the time and effort invested in making this manuscript more comprehensive and thorough. I only have two technical comments, and once they are addressed, I recommend acceptance of the manuscript.

1. If I have counted the number of spectra in Table 2 correctly, it ends up with 202. Section 4.1 says, "From this set, only 202 spectra are used for training the CIC algorithm" I assume this means that Section 3.1 (which is related to Table 2) uses the same 202 spectra for training. If so, I would like to suggest mentioning this in 3.1 such that readers will be aware that the same set of training spectra were used in both Sections 3.1 and 4.1.

Some text is added to better clarify this. The total number of TS spectra is reported in the table 2 caption note. In Section 4.1 a reference to Table 2 is added which should make clear that the spectra used in the TS are those reported in the Table. Another similar specification is provided at line 238 in Section 3.1.

2. Figure 10 in reply to my previous comments (acp-2021-97-AC1-supplement.pdf) is different from Figure 10 in the revised manuscript. The author should make sure to use the correct one in the final manuscript.

True. The final version of the figure is the one reported in the revised manuscript. The plot does not include the comparison with the MODIS MCD06COSP products since it is based on the combined used of visible and infrared channels. Our choice is also supported by the request of Reviewer #1 who suggested to avoid the usage of that specific MODIS product.